# ADDITIVE SEPARABLE GRAPHON MODELS

## ABSTRACT

The graphon function is fundamental to modeling exchangeable graphs, which form the basis for a wide variety of networks. In this paper, we use the additive separable model as a parsimonious representation of the graphon, capable of generating a low-rank connection probability matrix for network data. This model effectively addresses the well-known identification challenges associated with graphon functions. We develop an efficient estimation approach that leverages subgraph counts to estimate the low-rank connection matrix and uses interpolation to recover the graphon functions, achieving the minimax optimal estimation rate. We provide the convergence rate of our method, and validate its computational efficiency and estimation accuracy through comprehensive simulation studies.

## 1 INTRODUCTION

With advancements in data collection and analysis techniques, the modeling of network data has become increasingly prevalent. Examples of network data include brain networks (Maugis et al., 2020), co-authorship networks (Isfandyari-Moghaddam et al., 2023), and biological networks (Kamimoto et al., 2023), among others. A critical challenge in analyzing such data is understanding the underlying generative mechanisms, which are essential for tasks like studying dynamic evolution (Pensky, 2019), predicting links (Gao et al., 2016), and detecting communities (Jin et al., 2021). One effective modeling framework is that of exchangeable graphs, which represent networks as graphs where nodes correspond to objects and edges represent connections between them. The concept of exchangeability implies that the joint distribution of edges remains invariant under any permutation of the nodes. According to the Aldous-Hoover theorem, any exchangeable graph can be characterized by a graph function, commonly known as a graphon. The study of graphons has gained substantial attention in the literature due to their ability to tackle a wide array of challenges, including explaining the asymptotic normality of subgraphs (Bickel et al., 2011) and conducting hypothesis testing for the equivalence of two graphs (Maugis et al., 2020). Furthermore, the graphon model encompasses several widely used models as special cases, such as the stochastic block model (SBM) (Holland et al., 1983), the random dot product graph (RDPG) model (Young & Scheinerman, 2007), and the latent space model (Hoff et al., 2002).

The graphon is a symmetric, measurable bivariate function, which, without additional assumptions, is not directly estimable from a single observed network. However, by leveraging its eigenvalue-eigenfunction decomposition, we can impose a highly effective assumption: truncating the decomposition to retain terms with leading eigenvalues, similar to principal component analysis. A key benefit of this approach is that the resulting connection probability matrix $P$–the matrix formed when the graphon function is evaluated at the nodes–naturally inherits the same low rank as the truncated graphon.

Building on this insight, we propose the Additive Separable Graphon (ASG) models, which elegantly align the low-rank properties of both the graphon and the connection probability matrix. To estimate this new low-rank network model, we have developed a highly efficient method that harnesses well-established, scalable techniques for subgraph counting, enabling rapid and practical implementation. The core idea is intuitive: a matrix of rank $r$ can be decomposed into a sum of $r$ rank-1 matrices. By counting the number of $O(r)$ subgraphs, we extract the information corresponding to these $r$ rank-1 matrices. Solving the associated system of equations allows us to estimate each matrix individually. By combining these estimates, we reconstruct the final rank-$r$ matrix, which serves as our estimator for $P$. The graphon function $f$ is then estimated by utilizing sorting and

interpolation techniques based on these rank-1 matrix estimates. The graphon captures the limiting behavior of a graphon model, and our method leverages the low-rank structure of the graphon for estimation. Notably, our method is tuning parameter-free, as it does not require bandwidth or other adjustments, and it performs consistently well across various settings. On the other hand, estimating the connection probability matrix is important in practice and has attracted increasing attention, see for example, all of which focus on estimating the connection probability matrix rather than the graphon itself. In contrast, our method can estimate the graphon function and connection probability matrix simultaneously.

In the existing literature, graphon estimation methods can be broadly categorized into two groups: those focused on estimating the graphon itself, and those targeting the connection probability matrix $P$. For graphon estimation, Olhede & Wolfe (2014) approximate the graphon using blocks, treating it as a two-dimensional step function. Their method, however, requires finding the argmax of a permutation, and their implementation relies on a greedy search algorithm that can be computationally slow (see Section 4 for more details). Chan & Airoldi (2014) refine this by reordering nodes based on degree and applying total variation minimization, assuming the one-dimensional marginals of the graphon are strictly monotone. However, this assumption is restrictive, as it excludes SBM. The resulting $P$ from either of these approaches is generally not low-rank.

On the other hand, methods focused on estimating $P$ include Chatterjee (2015), who impose a low-rank assumption on $P$ and propose using Universal Singular-Value Thresholding (USVT), treating the adjacency matrix as a perturbed version of $P$. Zhang et al. (2017) apply neighborhood smoothing to estimate $P$, achieving near-minimax optimality. Gao et al. (2016) propose a minimax optimal combinatorial least-squares estimator, which is also adopted by Wu et al. (2024+) for non-exchangeable network. Despite these advances, due to identification issues, graphon functions cannot be directly recovered from the connecting probability matrix produced by these methods.

The structure of this paper is organized as follows: Section 2 introduces our parsimonious low-rank graphon model, termed the Additive Separable Graphon Model (ASG). Section 3 provides a comprehensive description of the estimation procedures and algorithms for both $r = 1$ and $r \geq 2$. In Section 4, we present simulation studies that demonstrate the advantages of the proposed method in terms of implementation speed and estimation accuracy. Section 5 offers a discussion and outlines potential directions for future research. Finally, an analysis of time complexity, a real data example, additional simulation results, an approach for selecting the rank $r$ when it is unknown, the proofs of the theoretical results, and the corresponding technical lemmas are provided in the appendix.

## 1.1 NOTATIONS

For a real number $x$, $\lfloor x \rfloor$ denotes the greatest integer less than or equal to $x$. For two positive real numbers $a$ and $b$, we define $a \vee b = \max(a, b)$ and $a \wedge b = \min(a, b)$. Let $\|A\|_F$ represent the Frobenius norm of a matrix $A$, and let $A_{ij}$ denote the element in the $i$-th row and $j$-th column. For two sequences of positive real numbers $a_n$ and $b_n$, we write $a_n = O(b_n)$ or $a_n \lesssim b_n$ if there exist positive constants $N$ and $C$ such that $\frac{a_n}{b_n} \leq C$ for all $n > N$. For two sequences of random variables $X_n$ and $Y_n$, we write $X_n = O_p(Y_n)$ if for any $\varepsilon > 0$, there is a constant $C_\varepsilon > 0$ such that $\sup_n \mathbb{P}(|X_n| \geq C_\varepsilon |Y_n|) < \varepsilon$.

## 2 ADDITIVE SEPARABLE GRAPHON MODEL

We consider a random graph $\mathcal{G} = (V, E)$ using the graphon formulation. Specifically, for $i = 1, \ldots, n$, where $n$ represents the size of the network, each node $i$ is associated with an independent and identically distributed (i.i.d.) random variable $U_i \sim \text{Uniform}(0, 1)$. The edges $E_{ij}$ are then independently drawn as $E_{ij} \sim \text{Bernoulli}(f(U_i, U_j))$ for $i < j$, where $f(\cdot, \cdot)$ is a symmetric, measurable function $f : [0, 1]^2 \to [0, 1]$ named the graphon. Additionally, we have $E_{ii} = 0$ and $E_{ij} = E_{ji}$ for $i > j$. Although we focus on undirected graphs without self-loops, our methods and theory can be readily extended to graphs with self-loops or directed graphs. Notably, many large-scale real-world networks exhibit low-rank characteristics, such as memberships or communities. The SBM (Holland et al., 1983) and the RDPG (Young & Scheinerman, 2007) are popular approaches for capturing unobserved heterogeneity in networks. For further discussion and real data examples, we refer to Athreya et al. (2018); Thibeault et al. (2024); Fortunato (2010).

To incorporate low-rank structure into graphon models, we propose a parsimonious model called the additive separable graphon model with rank $r$ (ASG($r$)), defined as follows:

$$f(U_i, U_j) = \sum_{k=1}^{r} \lambda_k G_k(U_i) G_k(U_j), \tag{1}$$

where $|\lambda_1| \geq |\lambda_2| \geq \cdots \geq |\lambda_r| > 0$, $G_k$ is a measurable function, $\int_0^1 G_k^2(u) \, du = 1$ for $k = 1, \ldots, r$, and $\int_0^1 G_k(u) G_l(u) \, du = 0$ for $k \neq l$. This can be viewed as a truncated eigen decomposition of the graphon, as suggested by the Hilbert-Schmidt theorem; see, for example, Szegedy (2011). Model (1) includes the aforementioned SBM and RDPG as special cases. For example, if the $G_k$ functions are step functions, model (1) reduces to an SBM with $r$ blocks. Moreover, if all $\lambda_k$ values are positive, our model simplifies to a rank $r$ RDPG. The introduction of low-rank structures in graphons not only enhances the ability of existing graphon models to capture real-world low-rank features of network data, but also offers computational advantages due to the additive separable structure. In this paper, we propose a novel, computationally efficient, and theoretically justified method to estimate the connection probabilities $\{f(U_i, U_j)\}_{i,j=1}^{n}$ and the full graphon function $f$. It is important to note that due to identification issues, effective approaches for estimating general graphon functions in polynomial time are rare in practice (Gao & Ma, 2021).

## 3 METHODOLOGY AND THEORY

Let $p_{ij} = f(U_i, U_j)$ be the connection probability between the $i$-th and $j$-th nodes, with $P = (p_{ij})_{i,j}$ as the connection probability matrix. While we focus on estimating $P$ and $f$ for $r = 1$ in Section 3.1, our method is readily extendable to cases where $r \geq 2$, which we provide in Section 3.2 for more detail.

### 3.1 ASG(1): ADDITIVE SEPARABLE GRAPHON WITH RANK-1

To clarify the motivation, we first consider the case where $r = 1$, i.e., $f(U_i, U_j) = \lambda_1 G_1(U_i) G_1(U_j)$. Assume, without loss of generality, that $\inf_{u \in [0,1]} G_1(u) \geq 0$; otherwise, we can replace $G_1(u)$ with $|G_1(u)|$. Note that the degree of the $i$-th node, $d_i = \sum_j E_{ij}$, satisfies

$$\frac{\mathbb{E}(d_i \mid U_i)}{n-1} = \frac{1}{n-1} \sum_{j \neq i} \int_0^1 f(U_i, U_j) \, dU_j = \lambda_1 G_1(U_i) \int_0^1 G_1(u) \, du, \tag{2}$$

which is proportional to $G_1(U_i)$. Moreover, by Lemma 2, we have

$$\sup_{i=1,\ldots,n} \frac{|d_i - \mathbb{E}(d_i \mid U_i)|}{n-1} = O_p(\sqrt{\log(n)/n}). \tag{3}$$

Therefore, $G_1(U_i)$ can be estimated by $\frac{d_i}{n-1}$, and consequently, $p_{ij}$ can be estimated by $\frac{d_i d_j}{(n-1)^2}$, both up to a multiplicative factor. Finally, we align with the sparsity of the graph $\mathcal{G}$ to provide a moment estimation of the multiplicative factor. We summarize the estimation procedure for $p_{ij}$ in Algorithm 1.

---

**Algorithm 1** Estimation procedure for $\{p_{ij}\}_{i,j=1}^{n}$ for ASG(1).

---

**Require:** The graph $\mathcal{G} = (V, E)$.
1: For $i = 1, \ldots, n$, let $d_i = \sum_{j: j \neq i} E_{ij}$.
2: Let $c_1 = \sum_{i,j: i \neq j} E_{ij} / \sum_{i,j: i \neq j} d_i d_j$.
3: For any $(i, j)$ pair, $i \neq j$, let the estimator of $p_{ij}$ be $\hat{p}_{ij} = 1 \wedge (c_1 d_i d_j)$.
4: Let $\hat{p}_{ii} = 0$ for $i = 1, \ldots, n$.
5: Output $\{\hat{p}_{ij}\}_{i,j=1}^{n}$.

---

Algorithm 1 is very simple, utilizing the low-rank setting $r = 1$. The time complexity of Algorithm 1 is $O(n^2)$, which is efficient considering that there are $O(n^2)$ values of $p_{ij}$ to be estimated. By comparison, SVD-based methods (e.g., Xu (2018)) typically require a time complexity of $O(n^3)$.

**Remark 1** (Comparison with power iteration method for estimating the connection probability matrix). *As an alternative method for computing the decomposition in our model, the power iteration approach (Mises & Pollaczek-Geiringer, 1929; Stoer et al., 1980) can be utilized. In our simulations, we compare it with our approach for estimating connection probability matrices. As shown in Tables 2 and 5, both methods perform comparably in dense regimes. However, our method demonstrates superior performance in the sparse regime, as evidenced in Table 3.*

We now present the theoretical results for the estimates $\hat{p}_{ij}$.

**Theorem 1.** *For ASG(1), assume that $\int_0^1 G_1(u)\,du > 0$. Applying Algorithm 1 to obtain the estimates $\hat{p}_{ij}$, we have $\sup_{i,j} |\hat{p}_{ij} - p_{ij}| = O_p(\sqrt{\log(n)/n})$.*

The assumption in Theorem 1 is mild and does not require the continuity of the function $G_1$. This flexibility allows our model to accommodate block structures, such as the SBM with a rank-1 connection probability matrix. Furthermore, the estimated connection probability matrix $\hat{P} = (\hat{p}_{ij})_{i,j}$ retains a rank of 1, consistent with the rank of $P$. Additionally, the result $\sup_{i,j} |\hat{p}_{ij} - p_{ij}| = O_p(\sqrt{\log(n)/n})$ implies the convergence $\|\hat{P} - P\|_F^2 / n^2 = O_p(\log(n)/n)$, a metric commonly used in the literature, such as by Zhang et al. (2017) and Gao et al. (2015). Estimating the graphon function $f(u, v)$ is generally challenging due to the identification issues caused by measure-preserving transformations (Borgs et al., 2015; Diaconis & Janson, 2007; Olhede & Wolfe, 2014). Consequently, many popular methods, including those in Gao et al. (2016) and Zhang et al. (2017), focus on estimating the connection probability matrix, as we present in Theorem 1.

In the low-rank case with $r = 1$, we can mitigate the non-identifiability issue by defining a canonical, monotonically non-decreasing graphon through rearrangement. Specifically, let

$$G_1^\dagger(u) = \inf\{t : \mu(G_1 \le t) \ge u\},$$

where $\mu(\cdot)$ denotes the Lebesgue measure. As shown in Barbarino et al. (2022), the function $G_1^\dagger(u)$ is the monotone rearrangement of $G_1(u)$, making it monotonically non-decreasing, left-continuous, and measure-preserving. Moreover, $G_1^\dagger(u)$ is continuous if $G_1(u)$ is continuous. Consequently, we can focus on the canonical graphon $f^\dagger(u, v) := \lambda_1 G_1^\dagger(u) G_1^\dagger(v)$.

To estimate $f^\dagger(u, v)$, we propose a degree sorting and interpolation method. Let $\sigma(k)$ denote the index $i$ corresponding to the $k$-th smallest value in the sequence $\{d_i\}_{i=1}^n$, i.e., $d_{\sigma(1)} \le d_{\sigma(2)} \le \cdots \le d_{\sigma(n)}$. Then, for any $(u, v) \in [0, 1]^2$, we define

$$\hat{f}^\dagger(u, v) := 1 \wedge (c_1 h(u) h(v)),$$

where

$$h(v) := \begin{cases} d_{\sigma(1)}, & \text{if } \lfloor v(n+1) \rfloor = 0, \\ d_{\sigma(\lfloor v(n+1) \rfloor)}(\lfloor v(n+1) \rfloor + 1 - v(n+1)) \\ \quad + d_{\sigma(\lfloor v(n+1) \rfloor + 1)}(v(n+1) - \lfloor v(n+1) \rfloor), & \text{if } 1 \le \lfloor v(n+1) \rfloor < n, \\ d_{\sigma(n)}, & \text{if } \lfloor v(n+1) \rfloor \ge n. \end{cases}$$

serves as an estimator of the graphon $f^\dagger(u, v)$. Intuitively, $U_{\sigma(i)}$ is close to $i/(n+1)$, allowing us to approximate the entire function using a piecewise linear approach. We then have the following result.

**Theorem 2.** *For ASG(1), assuming that $G_1(u)$ is Lipschitz continuous on the interval $[0, 1]$, i.e., there exists a constant $M > 0$ such that for any $u_1, u_2 \in [0, 1]$, $|G_1(u_1) - G_1(u_2)| \le M|u_1 - u_2|$. Then,*

$$\sup_{u,v \in [0,1]} |\hat{f}^\dagger(u, v) - f^\dagger(u, v)| \xrightarrow{a.s., L_2} 0, \text{and} = O_p(\sqrt{\log(n)/n}).$$

The estimation rate coincides with Chan & Airoldi (2014).

## 3.2 ASG(R): ADDITIVE SEPARABLE GRAPHON WITH RANK-R

For ASG(r), the connection probability $p_{ij} = f(U_i, U_j)$ is given by

$$p_{ij} = \sum_{k=1}^{r} \lambda_k G_k(U_i) G_k(U_j),$$

where $|\lambda_1| \geq |\lambda_2| \geq \cdots \geq |\lambda_r| > 0$, $\int_0^1 G_k^2(u)\, du = 1$ for $1 \leq k \leq r$, and $\int_0^1 G_i(u) G_j(u)\, du = 0$ for $1 \leq i \neq j \leq r$. To estimate $p_{ij}$, the key idea is to leverage additional subgraph counts to distinguish between the information derived from $G_k, 1 \leq k \leq r$. Since subgraph frequencies represent moments of the graphon, as suggested by Bickel et al. (2011), our approach can be seen as a form of moment estimation.

The study of subgraphs, often referred to as "motifs" in complex systems science, is not only theoretically significant (e.g., Maugis et al. (2020); Bravo-Hermsdorff et al. (2023); Ribeiro et al. (2021)) but also practically important (e.g., Milo et al. (2002); Dey et al. (2019); Yu et al. (2019)). For simplicity, in the case of ASG(r), we use subgraphs consisting of lines and cycles, as their expectations can be conveniently expressed using the graphon function. Moreover, selecting lines and cycles allows us to approximate them using paths with repeated nodes, which leads to a variant algorithm (Algorithm 3) that has the same time complexity as matrix multiplication (which is $O(n^{2.373})$).

Specifically, for $i = 1, \ldots, n$, let

$$L_i^{(1)} = \sum_{i_1} E_{i i_1}, \; L_i^{(a)} = \sum_{i_1, \cdots, i_a \text{ distinct}, i_k \neq i, 1 \leq k \leq a} E_{i i_1} \prod_{j=2}^{a} E_{i_{j-1} i_j} \text{ for } a \geq 2,$$

$$C_i^{(a)} = \sum_{i_1, \cdots, i_{a-1} \text{ distinct}, i_k \neq i, 1 \leq k \leq a-1} E_{i i_1} E_{i_{a-1} i} \prod_{j=2}^{a-1} E_{i_{j-1} i_j} \text{ for } a \geq 3. \tag{4}$$

In other words, $L_i^{(a)}$ represents the count of simple paths of length $a$ that have node $i$ as an endpoint, while $C_i^{(a)}$ represents the count of cycles of length $a$ with node $i$ as a point. By evaluating the expected counts of these subgraphs, we estimate the parameters $(\lambda_1, \cdots, \lambda_r, \int_0^1 G_1(u)\, du, \cdots, \int_0^1 G_r(u)\, du)$ by solving for $(\hat{\lambda}_1, \cdots, \hat{\lambda}_r, y_1, \cdots, y_r)$ in the following system of equations:

$$y_k \geq 0, \text{ for } 1 \leq k \leq r, |\hat{\lambda}_1| > \cdots > |\hat{\lambda}_r|,$$

$$\sum_{k=1}^{r} \hat{\lambda}_k^a = \frac{1}{\prod_{j=0}^{a-1}(n-j)} \sum_{i=1}^{n} C_i^{(a)} \text{ for } 3 \leq a \leq r+2, \tag{5}$$

$$\sum_{k=1}^{r} \hat{\lambda}_k^a y_k^2 = \frac{1}{\prod_{j=0}^{a}(n-j)} \sum_{i=1}^{n} L_i^{(a)} \text{ for } 1 \leq a \leq r. \tag{6}$$

As in (2), we express the conditional expectations as follows:

$$\frac{1}{\prod_{j=1}^{a}(n-j)} \mathbb{E}(L_i^{(a)} \mid U_i) = \sum_{k=1}^{r} \lambda_k^a G_k(U_i) \int_0^1 G_k(u)\, du \text{ for } 1 \leq a \leq r.$$

Then for every $1 \leq i \leq n$, we define the point-wise statistics $\hat{G}_k(U_i), 1 \leq k \leq r$, as the solution for $G_k(U_i), 1 \leq k \leq r$, in the following system of equations:

$$\frac{1}{\prod_{j=1}^{a}(n-j)} L_i^{(a)} = \sum_{k=1}^{r} \hat{\lambda}_k^a y_k G_k(U_i) \text{ for } 1 \leq a \leq r. \tag{7}$$

Additionally, we standardize $\hat{G}_k(U_i)$ as

$$\tilde{G}_k(U_i) = \frac{\hat{G}_k(U_i)}{\sqrt{\sum_{i=1}^{n} \hat{G}_k^2(U_i)/n}}. \tag{8}$$

We remark that this standardization step typically enhances performance in finite samples, benefiting both dense and sparse graphon settings. Then the estimated connection probabilities are then given by $\hat{p}_{ij} = \left[1 \wedge \left(0 \vee (\sum_{k=1}^{r} \hat{\lambda}_k \tilde{G}_k(U_i) \tilde{G}_k(U_j))\right)\right]$. The estimation procedure is summarized in Algorithm 2.

---

**Algorithm 2** Estimation procedure for $\{p_{ij}\}_{i,j=1}^{n}$ for ASG(r).

---

**Require:** The graph $\mathcal{G} = (V, E)$.
 1: For $i = 1, \ldots, n$, compute $L_i^{(a)}, 1 \le a \le r$ and $C_i^{(a)}, 3 \le a \le r+2$ defined in (4).
 2: Solve the system of equations in (6) to obtain $(\hat{\lambda}_1, \cdots, \hat{\lambda}_r, y_1, \cdots, y_r)$.
 3: For $i = 1, 2, \ldots, n$, compute the estimators $\hat{G}_1(U_i), \cdots, \hat{G}_r(U_i)$ from (7). Compute the standardized estimators $\tilde{G}_1(U_i), \cdots, \tilde{G}_r(U_i)$ from (8).
 4: For each pair $(i, j)$, where $i \ne j$, estimate $p_{ij}$ as $\hat{p}_{ij} = \left[1 \wedge \left(0 \vee (\sum_{k=1}^{r} \hat{\lambda}_k \tilde{G}_k(U_i) \tilde{G}_k(U_j))\right)\right]$. Set $\hat{p}_{ii} = 0$ for $i = 1, \ldots, n$.
 5: Output $\{\hat{p}_{ij}\}_{i,j=1}^{n}$.

---

**Remark 2.** *In Section A.1, we introduce a modified version of Algorithm 2 that retains all theoretical guarantees from Theorems 3 and 4 while achieving the time complexity of matrix multiplication, specifically $O(n^{2.373})$.*

**Remark 3** (Comparison with the spectral method for estimating the connection probability matrix)**.** *Spectral methods, such as USVT (Chatterjee, 2015), also estimate the connection probability matrix by computing eigenvalues and eigenvectors. However, our approach differs in several important ways. First, our motivation is fundamentally different: our primary goal is to estimate the graphon function itself, rather than simply the connection probability matrix. This distinction leads to a different methodology. While spectral methods rely on matrix spectral decomposition, our approach is based on subgraph counts, employing a moment-based technique that is a traditional and still evolving tool in statistical network analysis. Second, our method achieves the minimax rate for the mean squared error up to a logarithmic factor, without imposing smoothness assumptions on the graphon. In contrast, spectral methods typically require assumptions such as piecewise constant or Hölder-class smoothness of the graphon to derive convergence rates, as shown in Xu (2018). Lastly, for sparse graphons, our method empirically outperforms USVT, as demonstrated in Table 3, highlighting its advantage in handling networks with lower densities.*

We impose the following mild conditions for the consistency of $\hat{p}_{ij}$.

**Assumption 1.** *Assume that: (i) $|\lambda_1| > \cdots > |\lambda_r| > 0, \int_0^1 G_k^2(u)du = 1$, for $1 \le k \le r$, and $\int_0^1 G_i(u)G_j(u)du = 0$ for $1 \le i \ne j \le n$, (ii) $\int_0^1 G_k(u)du \ne 0$, for $1 \le k \le r$, (iii) there exists a constant $K > 0$ such that $\max_{1 \le k \le r} \sup_{u \in [0,1]} |G_k(u)| \le K$.*

Assumption 1 (i) is a standard condition ensuring the identifiability of the functions $G_k$. Intuitively, this condition is analogous to the restriction on eigengaps in the RDPG model (see, for example, Lyzinski et al. (2014)). Condition (ii) guarantees that the system of equations (7) has a unique solution, which is a similar requirement found in Bickel et al. (2011). Condition (iii) is mild and is typically satisfied by most graphon functions. It is worth noting that we do not require $G_k$'s to be piecewise smooth, which enhances the generality and applicability of our model in terms of estimating the connection probability matrix. We present the theoretical result for $\hat{p}_{ij}$ as follows.

**Theorem 3.** *For ASG(r), under Assumption 1, when $n$ is sufficiently large, there exists an open set $U \subset \mathbb{R}^{2r}$ containing the point $(\lambda_1, \cdots, \lambda_r, \int_0^1 G_1(u)\,du, \cdots, \int_0^1 G_r(u)\,du)$ such that, with probability 1, the system of equations in (6) has a unique solution within this region. Moreover, for $\hat{\lambda}_k, 1 \le k \le r, \hat{p}_{ij}$, we have $\max_{1 \le k \le r} |\hat{\lambda}_k - \lambda_k| = O_p(n^{-1/2})$, and $\sup_{i,j} |\hat{p}_{ij} - p_{ij}| = O_p(\sqrt{\log(n)/n})$.*

Importantly, Theorem 3 implies that $\|\hat{P} - P\|_F^2/n^2 = O_p(\log(n)/n)$, which matches the minimax rate (up to a logarithmic factor) that can be derived by following the proof of Theorem 1.1 in Gao et al. (2015). The estimation of graphon functions for ASG(r) presents more challenges than for ASG(1) due to the additive structure. We consider the following assumptions for estimating the graphon function of ASG(r):

**Assumption 2.** *Assume that: (i) At least one of $G_k, 1 \leq k \leq r$, is strictly monotonically increasing; and (ii) All $G_k, 1 \leq k \leq r$, are Lipschitz continuous with Lipschitz constant $M$.*

Assumption 2 serves as a technical condition that establishes an analogy to the "canonical form" for graphon functions of ASG(r). In this context, we refer to the monotone graphon as the reference marginal graphon. Chan & Airoldi (2014) proposed an alternative identification condition for graphon estimation, requiring the existence of a canonical form of the graphon that becomes strictly monotone after integrating out one of its arguments. It is also important to note that Assumption 2 is not necessary if the objective is to estimate the connection probability matrix rather than the entire graphon function.

Under Assumption 2, we proceed without loss of generality by assuming $G_1$ is the reference marginal graphon. We first sort the estimated pairs $(\hat{G}_1(U_i), \hat{G}_2(U_i), \cdots, \hat{G}_r(U_i))$ according to the first coordinate. Let $\gamma$ be a one-to-one permutation such that

$$\hat{G}_1(U_{\gamma(1)}) \leq \hat{G}_1(U_{\gamma(2)}) \leq \cdots \leq \hat{G}_1(U_{\gamma(n)}).$$

After sorting, we denote the reordered pairs as $(\hat{G}_1(U_{\gamma(i)}), \hat{G}_2(U_{\gamma(i)}), \cdots, \hat{G}_r(U_{\gamma(i)}))$. We then define the function

$$h_1(u) = \hat{G}_1(U_{\gamma(1)})I(u(n+1) < 1) + \hat{G}_1(U_{\gamma(n)})I(u(n+1) \geq n)$$

$$+ \sum_{k=1}^{n-1} \left( (k+1-u(n+1))\hat{G}_1(U_{\gamma(k)}) + (u(n+1)-k)\hat{G}_1(U_{\gamma(k+1)}) \right) I(\lfloor u(n+1) \rfloor = k)$$

as an estimate of the function $G_1$. For $G_k, k \geq 2,$, recognizing that $G_k$ is a function of $G_1$, we define:

$$h_k(u) = \hat{G}_k(U_{\gamma(1)})I(h_1(u) < \hat{G}_1(U_{\gamma(1)})) + \hat{G}_k(U_{\gamma(n)})I(h_1(u) \geq \hat{G}_1(U_{\gamma(n)}))$$

$$+ \sum_{k=1}^{n-1} \left( \frac{\hat{G}_1(U_{\gamma(k+1)}) - h_1(u)}{\hat{G}_1(U_{\gamma(k+1)}) - \hat{G}_1(U_{\gamma(k)})}\hat{G}_k(U_{\gamma(k)}) + \frac{h_1(u) - \hat{G}_1(U_{\gamma(k)})}{\hat{G}_1(U_{\gamma(k+1)}) - \hat{G}_1(U_{\gamma(k)})}\hat{G}_k(U_{\gamma(k+1)}) \right)$$

$$I(\hat{G}_1(U_{\gamma(k)}) \leq h_1(u) < \hat{G}_1(U_{\gamma(k+1)})).$$

Finally, we define

$$\hat{f}(u,v) := 1 \wedge \left(0 \vee \left(\sum_{k=1}^{r} \hat{\lambda}_k h_k(u) h_k(v)\right)\right) \tag{9}$$

as an estimate of the graphon $f(u,v)$.

Theorem 4 presents the theoretical result for this estimation. Since its proof follows directly from the proof of Theorem 2, we omit the details.

**Theorem 4.** *For ASG(r), under Assumptions 1 and 2, the estimated graphon given by (9) satisfies*

$$\sup_{u,v \in [0,1]} |\hat{f}(u,v) - f(u,v)| \overset{a.s.,L_2}{\longrightarrow} 0, and = O_p(\sqrt{\log(n)/n}).$$

The estimation rate coincides with Chan & Airoldi (2014).

**Remark 4.** *When $r$ is unknown, we can estimate it using a ratio-based method. Due to space limitations, we provide the detailed description in Appendix A.4.*

## 4 SIMULATIONS

In this section, we evaluate the effectiveness of our method through extensive simulation studies. For estimating the connection probability matrix $P$, we employ three metrics for assessment:

- Mean squared error (MSE) given by $\|\hat{P} - P\|_F^2/n^2$ (averaged with standard deviation) across 100 repetitions.

- Maximum error defined as $\max_{i \neq j} |\hat{p}_{ij} - p_{ij}|$ (averaged with standard deviation) across 100 repetitions.
- Average time cost measured in seconds.

The mean squared error is a standard metric commonly utilized in the literature. Additionally, we incorporate a stricter measure, namely the maximum error, to provide further insight into performance. To mitigate the impact of random fluctuations, we average both the MSE and the maximum error over 100 independent trials. For the estimation of the graphon function $f(u, v)$, we present visual representations of our estimated functions $G_k, 1 \leq k \leq r$ in Figure 1 to illustrate their performance. We generate networks from the seven graphons listed in Table 1, with the network size set to $n = 2000$.

| ID | Graphon $f(u, v)$ | Rank of $f(u, v)$ |
|---|---|---|
| 1 | 0.15 | 1 |
| 2 | $\frac{1.5}{(1+\exp(-u^2))(1+\exp(-v^2))}$ | 1 |
| 3 | $\frac{1}{5}\left(\tan\left(\frac{\pi}{2}u\right) + \frac{7}{6}\right)\left(\tan\left(\frac{\pi}{2}v\right) + \frac{7}{6}\right)$ | 1 |
| 4 | $0.95\exp(-3u)\exp(-3v) + 0.04(3u^2 - 5u + 1)(3v^2 - 5v + 1)$ | 2 |
| 5 | $\frac{1}{2}(\sin u \sin v + uv)$ | 2 |
| 6 | $0.05 + 0.15I(u < 0.4, v < 0.4) + 0.25I(u > 0.4, v > 0.4)$ | 2 |
| 7 | $0.1 + 0.75I(u, v < \frac{1}{3}) + 0.15I(\frac{1}{3} < u, v \leq \frac{2}{3})$ $+0.5I(u, v > \frac{2}{3})$ | 3 |

Table 1: List of Graphons. We estimate three rank-1 graphons using Algorithm 1, and four rank $\geq 2$ graphons using Algorithm 2.

For comparison, we include the universal singular value thresholding (USVT) method (Chatterjee, 2015) and the sort-and-smooth (SAS) method (Chan & Airoldi, 2014). Both algorithms demonstrate consistency and computational efficiency. Additionally, we compare our approach with the network histogram method (Olhede & Wolfe, 2014), and the neighborhood smoothing method proposed by Zhang et al. (2017). As discussed in Remark 1, we also include the power iteration method (Stoer et al., 1980). To streamline the discussion, we denote the following acronyms for these methods: N.S. for the method of Zhang et al. (2017), Nethist for Olhede & Wolfe (2014), USVT for Chatterjee (2015), SAS for Chan & Airoldi (2014), and P.I. for power iteration method. For a fair comparison, we additionally conducted simulations using the true $r$ for USVT (i.e., retaining only the first $r$ eigenpairs) when $r \geq 2$, which we denote as USVT($r$). For the aforementioned methods, we utilize the R functions provided by the respective authors with their default parameters. All results presented in this section were generated on an Apple M1 machine equipped with 16GB of RAM, running macOS Sonoma with R version 4.2.1.

**Remark 5.** *In Algorithm 3, we employ $\tilde{L}_i^{(a)}$ and $\tilde{C}_i^{(a)}$ as approximations for $L_i^{(a)}$ and $C_i^{(a)}$, enabling efficient computation. Though their equivalence has been proven in Theorem 5, applying certain corrections in practice can improve the finite-sample performance. Specifically, we let*

$$\check{L}_i^{(3)} = \tilde{L}_i^{(3)} - \tilde{L}_i^{(2)} - (\tilde{L}_i^{(1)})^2, \quad \check{C}_i^{(4)} = \tilde{C}_i^{(4)} - \tilde{L}_i^{(2)} - (\tilde{L}_i^{(1)})^2,$$

$$\check{C}_i^{(5)} = \tilde{C}_i^{(5)} - 2(\tilde{L}_i^{(1)} - 2)\tilde{C}_i^{(3)} - \frac{1}{n}\left(\sum_{k=1}^{n} \tilde{L}_k^{(1)}\right)\tilde{C}_i^{(3)} - 2\sum_{k=1}^{n} E_{ik}\tilde{C}_k^{(3)}$$

*and use $\check{L}_i^{(3)}, \check{C}_i^{(4)}, \check{C}_i^{(5)}$ to replace $\tilde{L}_i^{(3)}, \tilde{C}_i^{(4)}, \tilde{C}_i^{(5)}$ respectively in Algorithm 3. In fact, we have $\check{L}_i^{(3)} = L_i^{(3)}, \check{C}_i^{(4)} = C_i^{(4)}$, and $\check{C}_i^{(5)}$ is closer to $C_i^{(5)}$ compared to $\tilde{C}_i^{(5)}$.*

We present a comprehensive summary of the results for rank $\geq 2$ settings in Tables 2, with additional results for the rank 1 settings presented in the appendix. Our method exhibit good performance across various settings for both MSE and maximum error, achieving the best performance in the first and fourth settings. Additionally, our method demonstrates comparable speed to the SAS and P.I., while significantly outperforming all other methods in terms of computational efficiency.

Moreover, the accuracy of our method is generally on par with that of the USVT approach. Notably, under certain regular conditions, the USVT method is nearly minimax optimal in some scenarios regarding MSE, up to a logarithmic factor (see Theorems 2 and 4 in Xu (2018)). Therefore, it is

| ID | Method | MSE ($\times 10^{-4}$) | Std. dev of MSE ($\times 10^{-6}$) | Max. error ($\times 10^{-2}$) | Std dev.of max. error ($\times 10^{-3}$) | Run time (seconds) |
|---|---|---|---|---|---|---|
| 4 | Ours | 1.826 | 6.172 | 12.898 | 12.559 | 0.627 |
| | N.S. | 4.388 | 8.413 | 17.902 | 11.686 | 108.569 |
| | Nethist | 3.928 | 16.982 | 18.649 | 15.296 | 22.829 |
| | USVT | 7.617 | 17.079 | 13.637 | 13.036 | 10.064 |
| | SAS | 18.641 | 90.264 | 97.064 | 24.053 | 1.422 |
| | P.I. | 1.890 | 6.420 | 13.400 | 13.927 | 0.654 |
| | USVT(2) | 1.898 | 6.462 | 13.408 | 13.906 | 10.064 |
| 5 | Ours | 1.774 | 6.204 | 9.676 | 7.804 | 1.507 |
| | N.S. | 7.101 | 14.996 | 17.473 | 9.156 | 122.995 |
| | Nethist | 7.729 | 31.838 | 18.559 | 15.486 | 23.804 |
| | USVT | 1.769 | 6.078 | 9.661 | 7.871 | 13.684 |
| | SAS | 28.703 | 115.215 | 89.861 | 49.103 | 1.104 |
| | P.I. | 4.680 | 7.730 | 29.196 | 56.981 | 0.736 |
| | USVT(2) | 5.140 | 9.234 | 31.701 | 38.934 | 13.684 |
| 6 | Ours | 2.582 | 7.017 | 9.319 | 7.808 | 0.682 |
| | N.S. | 7.527 | 6.960 | 18.312 | 11.119 | 115.813 |
| | Nethist | 9.548 | 244.015 | 22.573 | 107.570 | 19.441 |
| | USVT | 2.383 | 6.082 | 10.052 | 8.343 | 11.169 |
| | SAS | 18.456 | 16.344 | 95.000 | 0.000 | 1.500 |
| | P.I. | 2.380 | 6.080 | 10.052 | 8.344 | 0.682 |
| | USVT(2) | 2.383 | 6.082 | 10.052 | 8.343 | 11.169 |
| 7 | Ours | 3.768 | 9.091 | 12.721 | 12.233 | 1.316 |
| | N.S. | 6.596 | 6.372 | 17.611 | 9.760 | 126.270 |
| | Nethist | 41.224 | 1574.245 | 59.567 | 96.247 | 20.238 |
| | USVT | 3.644 | 7.580 | 12.613 | 11.696 | 11.640 |
| | SAS | 20.552 | 22.529 | 90.000 | 0.000 | 1.701 |
| | P.I. | 3.640 | 7.580 | 12.613 | 11.696 | 1.005 |
| | USVT(3) | 3.644 | 7.580 | 12.613 | 11.696 | 11.640 |

Table 2: Results for rank $\geq 2$ graphons across 100 independent trials.

particularly encouraging that our method achieves accuracy comparable to USVT in practice while maintaining much lower computational complexity.

Importantly, our method operates without any tuning parameters, enhancing its robustness across various settings.

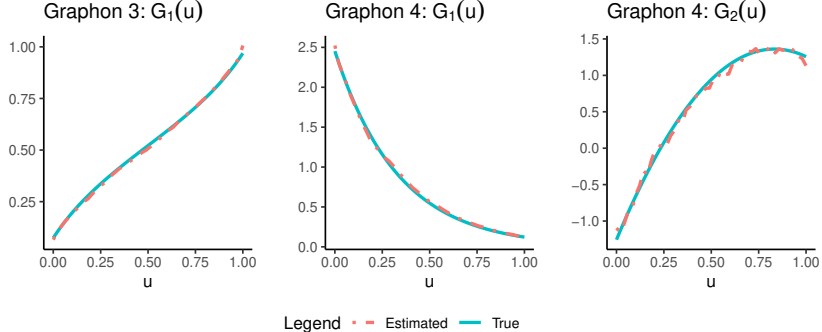

Figure 1: Estimation of graphons for the third and fourth settings.

To illustrate the accuracy of our graphon estimation, we applied the algorithms for estimating $G_1$ and $G_2$ as outlined at the end of Sections 3.1 and 3.2 to the third and fourth settings. The results presented in Figure 1 demonstrate that the estimated $G_1$ in the third setting aligns almost perfectly with the theoretical values. Furthermore, both estimated functions $G_1$ and $G_2$ closely match their theoretical counterparts, highlighting the effectiveness of our method. It is noteworthy that the function $G_2$ in the fourth setting is continuous but not monotonic.

In the remainder of this section, we evaluate the performance of our method in estimating connection probability matrices derived from sparse graphon models. Specifically, we consider the scenario where $E_{ij} \sim \text{Bernoulli}(\rho_n f(U_i, U_j))$, with $\rho_n \to 0$ indicating the degree of sparsity. We utilize the functions $f(x, y)$ from the previous 2nd, 3rd, 4th, and 5th settings, setting $\rho_n = n^{-1/2}$. For comparison, we include the same four methods as before, modifying the USVT method as suggested by Xu (2018) to ensure its adaptability to sparse settings. The results are summarized in Tables 3, which demonstrate that our method consistently outperforms the other five in terms of mean squared error (MSE).

Intuitively, our method is well-suited for handling sparse scenarios, as evident from equation (7), which incorporates the sparsity parameter $\rho_n$. We plan to explore further modifications of our proposed algorithm specifically tailored for sparse conditions, along with the corresponding detailed theoretical analysis, in future work.

| ID | Method | MSE ($\times 10^{-4}$) | Std. dev of MSE ($\times 10^{-6}$) | Max. error ($\times 10^{-2}$) | Std dev.of max. error ($\times 10^{-3}$) |
|----|--------|------|------|------|------|
|   | Ours | 0.115 | 0.401 | 2.560 | 3.367 |
|   | N.S. | 61.897 | 249.576 | 99.161 | 0.002 |
| 2 | Nethist | 0.391 | 1.536 | 2.112 | 2.926 |
|   | USVT | 0.352 | 2.524 | 1.790 | 0.017 |
|   | SAS | 0.946 | 5.704 | 99.16 | 0.013 |
|   | P.I. | 0.132 | 0.490 | 2.951 | 4.110 |
|   | Ours | 0.075 | 0.284 | 3.079 | 4.494 |
|   | N.S. | 40.084 | 119.792 | 99.968 | 0.093 |
| 3 | Nethist | 0.249 | 1.056 | 2.840 | 14.842 |
|   | USVT | 0.314 | 0.952 | 2.093 | 0.039 |
|   | SAS | 0.200 | 2.163 | 99.956 | 0.426 |
|   | P.I. | 0.099 | 0.445 | 4.255 | 6.704 |
|   | Ours | 0.043 | 0.398 | 3.840 | 8.423 |
|   | N.S. | 14.573 | 43.039 | 99.973 | 0.047 |
| 4 | Nethist | 0.111 | 0.642 | 2.485 | 12.126 |
|   | USVT | 0.102 | 0.648 | 2.202 | 0.075 |
|   | SAS | 0.078 | 0.945 | 89.227 | 253.955 |
|   | P.I. | 0.115 | 2.704 | 33.130 | 158.566 |
|   | Ours | 0.071 | 0.330 | 2.910 | 4.576 |
|   | N.S. | 34.490 | 115.091 | 99.993 | 0.039 |
| 5 | Nethist | 0.229 | 0.948 | 3.001 | 20.765 |
|   | USVT | 0.294 | 0.971 | 1.906 | 0.023 |
|   | SAS | 0.118 | 0.907 | 75.367 | 325.895 |
|   | P.I. | 0.170 | 4.392 | 21.174 | 101.657 |

Table 3: Results for sparse graphons characterized by $\rho_n = n^{-1/2}$. Our method consistently performs best in terms of MSE.

## 5 DISCUSSION

In this paper, we present an effective and efficient estimation method for the additive separable graphon (ASG) model based on subgraph counts. We provide theoretical justifications for the methods applied to $ASG(r)$ with fixed $r$, and evaluate their performance through simulation studies.

There are several promising directions for future research. In our simulations, we found that the performance of our method in sparse graphons is competitive. Therefore, investigating the convergence rate as well as optimality of our method in the context of sparse graphons would be a valuable next step. Additionally, exploring the selection of "optimal" subgraphs offers another important research avenue. Finally, it remains an open question whether our method can be extended to cases where $r$ diverges with $n$.

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

# A APPENDIX

## A.1 A VARIANT ALGORITHM AND TIME COMPLEXITY

The primary computational complexity of Algorithm 2 arises from counting lines and cycles within the graph. Notably, counting paths that allow repeated nodes is considerably simpler than counting simple paths (where nodes cannot repeat), as the former can be achieved via matrix multiplication with a complexity of $O(n^{2.373})$, see for example, Williams (2012). Motivated by this observation, we define paths that permit node repetition and propose a variant algorithm accordingly.

For $i = 1, \ldots, n$, define the lines and cycles *allowing repeated nodes* as

$$\tilde{L}_i^{(1)} = \sum_{i_1} E_{ii_1}, \ \tilde{L}_i^{(a)} = \sum_{i_1, \cdots, i_a} E_{ii_1} \prod_{j=2}^{a} E_{i_{j-1}i_j} \text{ for } a \geq 2,$$

$$\tilde{C}_i^{(a)} = \sum_{i_1, \cdots, i_{a-1}} E_{ii_1} E_{i_{a-1}i} \prod_{j=2}^{a-1} E_{i_{j-1}i_j} \text{ for } a \geq 3.$$

$\tilde{L}_i^{(a)}, \tilde{C}_i^{(a)}$ can be computed efficiently. Specifically, let $E^a$ denote the $a_{th}$ power of the adjacency matrix $E$, then we have $\tilde{L}_i^{(a)} = \sum_{j \neq i}(E^a)_{ij}, \tilde{C}_i^{(a)} = (E^a)_{ii}$. The variant algorithm (Algorithm 3) uses $\tilde{L}_i^{(a)}, \tilde{C}_i^{(a)}$ instead of $L_i^{(a)}, C_i^{(a)}$.

---

**Algorithm 3** Fast estimation procedure for $\{p_{ij}\}_{i,j=1}^n$ for ASG(r).

---

**Require:** The graph $\mathcal{G} = (V, E)$.
1: For $i = 1, \ldots, n$, let $\tilde{L}_i^{(a)} = \sum_{j \neq i}(E^a)_{ij}, 1 \leq a \leq r, \tilde{C}_i^{(a)} = (E^a)_{ii}, 3 \leq a \leq r + 2$.
2: Set $L_i^{(a)} = \tilde{L}_i^{(a)}, C_i^{(a)} = \tilde{C}_i^{(a)}$.
3: Follow from Line 2 of Algorithm 2 to estimate $\{\hat{p}_{ij}\}_{i,j=1}^n$.
4: Output $\{\hat{p}_{ij}\}_{i,j=1}^n$.

---

**Remark 6** (Time complexity of Algorithm 3). *Since all $\tilde{L}_i^{(a)}$ and $\tilde{C}_i^{(a)}$ for $1 \leq i \leq n$ and $1 \leq a \leq r$ can be computed using matrix multiplication, which has a time complexity of $O(n^{2.373})$, it directly follows that the overall time complexity of Algorithm 3 is also $O(n^{2.373})$.*

To analyze the theoretical properties, we present a key lemma showing that $\tilde{L}_i^{(a)}$ and $L_i^{(a)}$ (as well as $\tilde{C}_i^{(a)}$ and $C_i^{(a)}$) are sufficiently close, such that their differences do not impact the results of Theorem 3 and Theorem 4.

**Lemma 1.** *For ASG(r), under the assumptions of Theorem 3, we have*

$$\max_{1 \leq i \leq n} \max_{1 \leq a \leq r} \left| \frac{1}{\prod_{j=1}^a (n-j)} (\tilde{L}_i^{(a)} - L_i^{(a)}) \right| = o_p \left( \frac{1}{\sqrt{n}} \right),$$

$$\max_{1 \leq i \leq n} \max_{3 \leq a \leq r+2} \left| \frac{1}{\prod_{j=1}^{a-1} (n-j)} (\tilde{C}_i^{(a)} - C_i^{(a)}) \right| = o_p \left( \frac{1}{\sqrt{n}} \right).$$

With Lemma 1 established, it follows straightforwardly that the following theorem holds.

**Theorem 5.** *Theorem 3 and Theorem 4 remain valid when the fast estimation procedure described in Algorithm 3 is applied.*

## A.2 A REAL DATA EXAMPLE

To demonstrate the effectiveness of our method, we applied it to a real data example of contacts in a primary school. The data is collected by the SocioPatterns project[1] with active RFID devices, which generate a new data record every 20 seconds capturing information from the preceding 20 seconds.

---

[1] http://www.sociopatterns.org

Figure 2: Estimated connection probability matrix for the real data example.

Specifically, on October 1st, 2009, from 8:40 to 17:18, contact data were collected for a total of 236 individuals, with a total of 60623 records. We use these data to construct a undirected simple graph and use our method to estimate the underlining graphon structure. Specifically, let $E$ denote the contact matrix, i.e.,

$$E_{kl} = \begin{cases} 1 & \text{individuals } k \text{ and } l \text{ contacted at least once,} \\ 0 & \text{otherwise,} \end{cases}$$

Firstly, we select the rank $r$ by our Algorithm 3 with the threshold $\tau = 0.2$. The results, as shown in Table 4, leads us to select $r = 4$.

| Rank $r$ | $\hat{\lambda}_1$ | $\hat{\lambda}_2$ | $\hat{\lambda}_3$ | $\hat{\lambda}_4$ | $\hat{\lambda}_5$ |
|---|---|---|---|---|---|
| 2 | 0.264 | 0.159 | | | |
| 3 | 0.266 | 0.146 | 0.0593 | | |
| 4 | 0.271 | 0.118 | 0.118 | $-0.0992$ | |
| 5 | 0.272 | 0.117 | 0.0813 | $-0.0423$ | $-0.00721$ |

Table 4: Estimated eigenvalues from Algorithm 3 with respect to different choice of rank $r$.

Subsequently, we estimated the connection probability matrix using Algorithm 2, and the resulting heatmap is depicted in Figure 2. The smoothness of the heatmap is consistent with expectations for real-world in-person interaction scenarios.

Assuming Assumption 2, we estimated the graphon function of the network, taking $G_1$ as the reference marginal graphon. The estimated functions $h_1, \cdots, h_4$ are plotted in Figure 3. Then for any $(u, v) \in [0, 1]^2$, the estimated value of graphon function $\hat{f}(u, v)$ can be obtained from equation 9.

### A.3 RESULTS FOR RANK-1 SETTINGS

We present the results for rank-1 graphons in Table 5.

### A.4 SELECTING r WHEN IT IS UNKNOWN

In this section, we propose a method for selecting $r$ when it is unknown. Since $\hat{r}$ approximates $r$ by Theorem 3, we can start estimating from $r = 1$ and incrementally increase $r$. When $|\hat{\lambda}_k|$ is

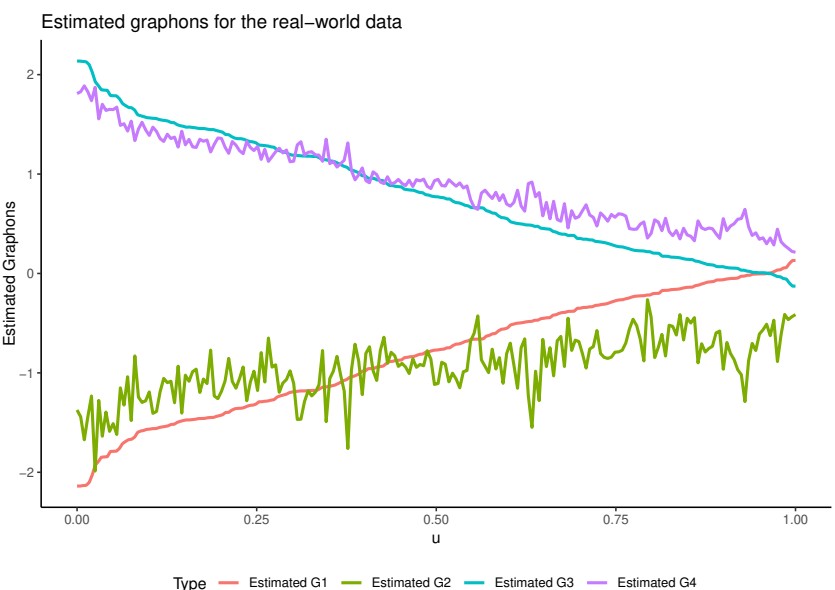

Figure 3: Estimated graphons for the real data example.

| ID | Method | MSE ($\times 10^{-4}$) | Std. dev of MSE ($\times 10^{-6}$) | Max. error ($\times 10^{-2}$) | Std dev.of max. error ($\times 10^{-3}$) | Run time (seconds) |
|---|---|---|---|---|---|---|
|   | Ours | 1.275 | 3.871 | 5.817 | 4.955 | 0.121 |
|   | N.S. | 7.853 | 5.175 | 16.749 | 9.849 | 115.076 |
| 1 | Nethist | 4.237 | 8.330 | 5.980 | 11.474 | 16.705 |
|   | USVT | 1.282 | 3.863 | 5.837 | 4.994 | 13.587 |
|   | SAS | 19.120 | 16.865 | 85.000 | 0.000 | 1.273 |
|   | P.I. | 1.280 | 3.860 | 5.837 | 4.994 | 0.304 |
|   | Ours | 2.452 | 7.806 | 8.114 | 5.819 | 0.539 |
|   | N.S. | 12.033 | 9.750 | 17.617 | 8.275 | 115.757 |
| 2 | Nethist | 9.867 | 24.220 | 16.962 | 44.037 | 16.744 |
|   | USVT | 2.403 | 7.593 | 7.977 | 5.740 | 14.629 |
|   | SAS | 39.888 | 29.339 | 78.134 | 37.486 | 1.250 |
|   | P.I. | 2.400 | 7.590 | 7.977 | 5.740 | 0.274 |
|   | Ours | 1.973 | 6.794 | 10.163 | 8.537 | 0.259 |
|   | N.S. | 8.337 | 14.186 | 17.329 | 8.566 | 114.694 |
| 3 | Nethist | 7.942 | 27.962 | 17.094 | 10.368 | 20.288 |
|   | USVT | 1.919 | 6.530 | 9.395 | 7.146 | 13.758 |
|   | SAS | 26.987 | 77.248 | 94.849 | 20.701 | 1.241 |
|   | P.I. | 1.920 | 6.530 | 9.395 | 7.146 | 0.328 |

Table 5: Results for rank-1 graphons across 100 independent trials.

significantly larger than 0, but $|\hat{\lambda}_i|, i \geq k+1$ are close to 0, we select $r = k$. The detailed selection procedure is summarized in Algorithm 4.

---

**Algorithm 4** Selection procedure for $r$.

---

**Require:** The graph $\mathcal{G} = (V, E)$, threshold $\tau$.

1: For $i = 1, \ldots, n$, compute $\tilde{C}_i^{(3)}$. Set $k = 1$.
2: For $i = 1, \ldots, n$, compute $\tilde{C}_i^{(k+3)}$.
3: Solve the system of equations in (5) with $3 \leq a \leq k+3$ and $r = k+1$ to obtain $(\hat{\lambda}_1, \cdots, \hat{\lambda}_{k+1})$.
4: If $\left| \frac{\hat{\lambda}_{k+1}}{\hat{\lambda}_k} \right| \leq \tau$, choose $r = k$ and output $r$.
5: Set $k = k+1$ and go back to Line 2.
6: Output $r$.

---

We apply Algorithm 4 to select $r$ for the third and sixth settings in Table 1, with $\tau = 0.2$. The results are summarized in Table 6. From the results, it can be observed that Algorithm 4 is effective in most cases.

| ID | True $r$ | Estimated $r$ | | | |
|----|----------|---|---|---|---|
|    |          | 1 | 2 | 3 | $\geq 4$ |
| 3  | 1        | 100 | 0 | 0 | 0 |
| 6  | 2        | 0 | 92 | 0 | 8 |

Table 6: Results for selection of $r$ for the third and sixth settings across 100 independent trials.

## A.5 Proofs

*Proof of Theorem 1.* By (2) and (3), we have

$$\sup_i \left| G_1(U_i) - \frac{1}{c(n-1)} d_i \right| = O_p(\sqrt{\log(n)/n}) \tag{10}$$

where $c = \lambda_1 \int_0^1 G_1(u) du$.

By the property of U statistics (see for example, Theorem 4.2.1 in Korolyuk (2013)), we have

$$\frac{1}{n(n-1)} \sum_{i,j:i\neq j} f(U_i, U_j) = \mathbb{E}f(U_i, U_j) + O_p(n^{-1/2}). \tag{11}$$

Moreover, note that

$$\mathbb{E}\left( \left( \frac{1}{n(n-1)} \sum_{i,j:i\neq j} E_{ij} - \frac{1}{n(n-1)} \sum_{i,j:i\neq j} f(U_i, U_j) \right)^2 \middle| U_1, \cdots, U_n \right) \tag{12}$$

$$\lesssim \frac{1}{n^4} \sum_{i_1,i_2,j_1,j_2} \mathbb{E}\left( (E_{i_1 j_1} - f(U_{i_1}, U_{j_1}))(E_{i_2 j_2} - f(U_{i_2}, U_{j_2}) \middle| U_1, \cdots, U_n \right) \tag{13}$$

$$\lesssim \frac{1}{n^4} \sum_{i_1,i_2} \mathbb{E}\left( (E_{i_1 j_1} - f(U_{i_1}, U_{j_1}))^2 \middle| U_1, \cdots, U_n \right) = O\left( \frac{1}{n^2} \right), \tag{14}$$

where the second inequality follows from the fact that the terms are nonzero only when $i_1 = i_2, j_1 = j_2$, and the last equality is due to the boundedness of each term. By combining (11) and (12), we have that

$$\frac{1}{n(n-1)} \sum_{i,j:i\neq j} E_{ij} = \lambda_1 \left( \int_0^1 G_1(u) du \right)^2 + O_p(n^{-1/2}). \tag{15}$$

Similarly,

$$\frac{1}{n(n-1)^3} \sum_{i,j:i\neq j} d_i d_j = \lambda_1^2 \left( \int_0^1 G_1(u) du \right)^4 + O_p(n^{-1/2}). \tag{16}$$

Combining (15) and (16), we obtain that

$$(n-1)^2 \frac{\sum_{i,j:i\neq j} E_{ij}}{\sum_{i,j:i\neq j} d_i d_j} = \frac{1}{\lambda_1 \left(\int_0^1 G_1(u)du\right)^2} + O_p(n^{-1/2}).$$

Hence,

$$\sup_i \left| G_1(U_i) - \sqrt{\frac{\sum_{i,j:i\neq j} E_{ij}}{\sum_{i,j:i\neq j} d_i d_j}} \frac{d_i}{\sqrt{\lambda_1}} \right| \tag{17}$$

$$\leq \sup_i \left| G_1(U_i) - \frac{1}{c(n-1)} d_i \right| + \sup_i \left| \sqrt{\frac{\sum_{i,j:i\neq j} E_{ij}}{\sum_{i,j:i\neq j} d_i d_j}} \frac{d_i}{\sqrt{\lambda_1}} - \frac{1}{c(n-1)} d_i \right| \tag{18}$$

$$\leq \sup_i \left| G_1(U_i) - \frac{1}{c(n-1)} d_i \right| + \left| \sqrt{(n-1)^2 \frac{\sum_{i,j:i\neq j} E_{ij}}{\sum_{i,j:i\neq j} d_i d_j}} \frac{1}{\sqrt{\lambda_1}} - \frac{1}{\lambda_1 \int_0^1 G_1(u)du} \right| \tag{19}$$

$$= O_p(\sqrt{\log(n)/n}). \tag{20}$$

By the definition of graphon function, $\sup_{u_1,u_2\in[0,1]} \lambda_1 G_1(u_1)G_1(u_2) \leq 1$. As a result, $\sup_{u\in[0,1]} \sqrt{\lambda_1} G_1(u) \leq 1$. Then for $c_1 = \sum_{i,j:i\neq j} E_{ij} / \sum_{i,j:i\neq j} d_i d_j$, we have

$$\sup_{i,j} |\hat{p}_{ij} - p_{ij}| \leq \sup_{i,j} |c_1 d_i d_j - \lambda_1 G_1(U_i) G_1(U_j)|$$

$$\leq \sup_{i,j} |\sqrt{c_1} d_i - \sqrt{\lambda_1} G_1(U_i)| \sqrt{c_1} d_j + \sup_{i,j} |\sqrt{c_1} d_j - \sqrt{\lambda_1} G_1(U_j)| \sqrt{\lambda_1} G_1(U_i)$$

$$\leq \sup_{i,j} |\sqrt{c_1} d_i - \sqrt{\lambda_1} G_1(U_i)| |\sqrt{c_1} d_j - \sqrt{\lambda_1} G_1(U_j)|$$

$$+ 2 \sup_{i,j} |\sqrt{c_1} d_j - \sqrt{\lambda_1} G_1(U_j)| \sqrt{\lambda_1} G_1(U_i)$$

$$= O_p(\sqrt{\log(n)/n}).$$

$\square$

*Proof of theorem 2.* It suffices to show that

$$\sup_{u\in[0,1]} \left| G_1^\dagger(u) - \frac{1}{(n-1)\lambda_1 \int_0^1 G_1(v)dv} h(u) \right| \overset{a.s.}{\to} 0, \tag{21}$$

$$\sup_{u\in[0,1]} \left| G_1^\dagger(u) - \frac{1}{(n-1)\lambda_1 \int_0^1 G_1(v)dv} h(u) \right| = O_p(\sqrt{\log(n)/n}), \tag{22}$$

then the theorem holds following the similar proof of Theorem 1 via following from (10) to (17) to replace $(n-1)\lambda_1 \int_0^1 G_1(v)dv$ by $\sqrt{\sum_{i,j:i\neq j} E_{ij}/(\lambda_1 \sum_{i,j:i\neq j} d_i d_j)}$, and via modifying the argument on taking maximum over all $U_i$ to taking supreme over all $u \in [0,1]$. To show (21) and (22), we consider the following two steps.

**(Step 1.)** In this step, we prove that

$$\sup_{u\in\{1,2,\cdots,n\}} |h(u/(n+1)) / ((n-1)\lambda_1 \int_0^1 G_1(v)dv) - G_1^\dagger(u/(n+1))| \overset{a.s.}{\to} 0,$$

and

$$\sup_{u\in\{1,2,\cdots,n\}} |h(u/(n+1)) / ((n-1)\lambda_1 \int_0^1 G_1(v)dv) - G_1^\dagger(u/(n+1))| = O_p(\sqrt{\log(n)/n}).$$

Let $U_{(1)}, \cdots, U_{(n)}$ denote the rearrangement of $U_1, \cdots, U_n \overset{i.i.d.}{\sim} Uniform(0,1)$ such that $U_{(1)} \leq \cdots \leq U_{(n)}$. By Lemma 3, we have $\sup_{i=1,\cdots,n} |U_{(i)} - i/(n+1)| \overset{a.s.}{\to} 0$. By Kawohl (2006)

(chapter II.2), the rearrangement function $G_1^\dagger$ is Lipschitz continuous with constant $M$ as long as $G_1$ is Lipschitz continuous with constant $M$. As a consequence,

$$\sup_{i=1,\cdots,n} |G_1^\dagger(U_{(i)}) - G_1^\dagger(i/(n+1))| \le M \sup_{i=1,\cdots,n} |U_{(i)} - i/(n+1)| \overset{a.s.}{\to} 0. \tag{23}$$

Moreover, via using the proof of Lemma 1 in Chan & Airoldi (2014), $\sup_{i=1,\cdots,n} |U_{(i)} - i/(n+1)| = O_p(\sqrt{\log(n)/n})$, which also shows that

$$\sup_{i=1,\cdots,n} |G_1^\dagger(U_{(i)}) - G_1^\dagger(i/(n+1))| = O_p(\sqrt{\log(n)/n}). \tag{24}$$

By definition, for $i = 1, \cdots, n$, $h(i/(n+1)) = d_{\sigma(i)}$. By (10) (more precisely, the similar argument of (10) applied to $G^\dagger$), (23) and Lemma 4, we have that

$$\sup_{i\in\{1,2,\cdots,n\}} \left| h\left(\frac{i}{n+1}\right) \frac{1}{(n-1)\lambda_1 \int_0^1 G_1(v)dv} - G_1^\dagger\left(\frac{i}{n+1}\right) \right| \overset{a.s.}{\to} 0.$$

Similarly, via (10), (24) and Lemma 4 we have

$$\sup_{i\in\{1,2,\cdots,n\}} \left| h\left(\frac{i}{n+1}\right) \frac{1}{(n-1)\lambda_1 \int_0^1 G_1(v)dv} - G_1^\dagger\left(\frac{i}{n+1}\right) \right| = O_p(\sqrt{\log(n)/n})$$

**(Step 2.)** In this step, we prove (21). We note that

$$\sup_{u\in[0,1/(n+1)]} \left| G_1^\dagger(u) - \frac{1}{(n-1)\lambda_1 \int_0^1 G_1(v)dv} h(u) \right|$$

$$\le \left| G_1^\dagger\left(\frac{1}{n+1}\right) - \frac{h(1/(n+1))}{(n-1)\lambda_1 \int_0^1 G_1(v)dv} \right| + \sup_{u\in[0,1/(n+1)]} \left| G_1^\dagger\left(\frac{1}{n+1}\right) - G_1^\dagger(u) \right|$$

$$\le \left| G_1^\dagger\left(\frac{1}{n+1}\right) - \frac{h(1/(n+1))}{(n-1)\lambda_1 \int_0^1 G_1(v)dv} \right| + \frac{M}{n+1} \overset{a.s.}{\to} 0, \text{and} = O_p(\sqrt{\log(n)/n})$$

Similarly, we have

$$\sup_{u\in[n/(n+1),1]} \left| G_1^\dagger(u) - \frac{1}{(n-1)\lambda_1 \int_0^1 G_1(v)dv} h(u) \right| \overset{a.s.}{\to} 0, \text{and} = O_p(\sqrt{\log(n)/n}).$$

For $u \in (1/(n+1), n/(n+1))$, let $k = \lfloor u(n+1) \rfloor$, then

$$\left| G_1^\dagger(u) - \frac{h(u)}{(n-1)\lambda_1 \int_0^1 G_1(v)dv} \right| \le (k+1-u(n+1)) \left| G_1^\dagger(u) - \frac{d_{\sigma(k)}}{(n-1)\lambda_1 \int_0^1 G_1(v)dv} \right|$$

$$+ (u(n+1)-k) \left| G_1^\dagger(u) - \frac{d_{\sigma(k+1)}}{(n-1)\lambda_1 \int_0^1 G_1(v)dv} \right| \le (k+1-u(n+1))|G_1^\dagger(u) - G_1^\dagger(k/(n+1))|$$

$$+ (k+1-u(n+1)) \left| G_1^\dagger(k/(n+1)) - \frac{h(k/(n+1))}{(n-1)\lambda_1 \int_0^1 G_1(v)dv} \right|$$

$$+ (u(n+1)-k) \left| G_1^\dagger((k+1)/(n+1)) - \frac{h((k+1)/(n+1))}{(n-1)\lambda_1 \int_0^1 G_1(v)dv} \right|$$

$$+ (u(n+1)-k)|G_1^\dagger(u) - G_1^\dagger((k+1)/(n+1))|$$

$$\le \frac{M}{n+1} + \sup_{i\in\{1,2,\cdots,n\}} \left| h\left(\frac{i}{n+1}\right) \frac{1}{(n-1)\lambda_1 \int_0^1 G_1(v)dv} - G_1^\dagger\left(\frac{i}{n+1}\right) \right|.$$

Therefore, by the result from **(Step 1)**,

$$\sup_{u\in[1/(n+1),n/(n+1)]} \left| G_1^\dagger(u) - \frac{1}{(n-1)\lambda_1 \int_0^1 G_1(v)dv} h(u) \right| \overset{a.s.}{\to} 0, \text{and} = O_p(\sqrt{\log(n)/n}).$$

The proof is then complete. $\qquad\square$

*Proof of Theorem 3.* Without loss of generality, we assume that $\int_0^1 G_k(u)du \geq 0, 1 \leq k \leq r$, because we can replace $G_k$ by $-G_k$ if $\int_0^1 G_k(u)du \leq 0$.

For $i = 1, \cdots, n$, recall that

$$L_i^{(1)} = \sum_{i_1} A_{ii_1},$$

$$L_i^{(a)} = \sum_{i_1,\cdots,i_a \text{ distinct}, i_k \neq i, 1 \leq k \leq a} E_{ii_1} \prod_{j=2}^{a} E_{i_{j-1}i_j} \text{ for } a \geq 2,$$

$$C_i^{(a)} = \sum_{i_1,\cdots,i_{a-1} \text{ distinct}, i_k \neq i, 1 \leq k \leq a-1} E_{ii_1} E_{i_{a-1}i} \prod_{j=2}^{a-1} E_{i_{j-1}i_j} \text{ for } a \geq 3.$$

Note that $\mathbb{P}(E_{ij} = 1|U_i, U_j) = \sum_{k=1}^{r} \lambda_k G_k(U_i)G_k(U_j)$ and $\int_0^1 G_i^2(u)du = 1$ for $1 \leq i \leq r$, we then have

$$\frac{1}{\prod_{j=1}^{a}(n-j)}\mathbb{E}(L_i^{(a)} \mid U_i) = \sum_{k=1}^{r} \lambda_k^a G_k(U_i) \int_0^1 G_k(u)\,du \text{ for } 1 \leq a \leq r,$$

$$\frac{1}{\prod_{j=1}^{a-1}(n-j)}\mathbb{E}(C_i^{(a)} \mid U_i) = \sum_{k=1}^{r} \lambda_k^a G_k^2(U_i) \text{ for } 3 \leq a \leq r+2. \tag{25}$$

We show the theorem via two steps.

**(Step 1.)** We first prove that $\max_{1 \leq k \leq r} |\hat{\lambda}_k - \lambda_k| = O_p(n^{-1/2})$, $\max_{1 \leq k \leq r} \left| y_k - \int_0^1 G_k(u)du \right| = O_p(n^{-1/2})$.

By (25), we have

$$\frac{1}{\prod_{j=1}^{a}(n-j)}\mathbb{E}(L_i^{(a)}) = \sum_{k=1}^{r} \lambda_k^a \left( \int_0^1 G_k(u)\,du \right)^2 \text{ for } 1 \leq a \leq r,$$

$$\frac{1}{\prod_{j=1}^{a-1}(n-j)}\mathbb{E}(C_i^{(a)}) = \sum_{k=1}^{r} \lambda_k^a \text{ for } 3 \leq a \leq r+2. \tag{26}$$

Moreover, by implicit function theorem, the system of equations (26) in terms of $\lambda_k, \left( \int_0^1 G_k(u)du \right)^2, 1 \leq k \leq r$, has a unique solution if

$$\begin{vmatrix} \lambda_1^2 & \lambda_2^2 & \cdots & \lambda_r^2 \\ \vdots & \cdots & \cdots & \vdots \\ \lambda_1^{r+1} & \lambda_2^{r+1} & \cdots & \lambda_r^{r+1} \end{vmatrix} \neq 0, \begin{vmatrix} \lambda_1 & \lambda_2 & \cdots & \lambda_r \\ \vdots & \cdots & \cdots & \vdots \\ \lambda_1^r & \lambda_2^r & \cdots & \lambda_r^r \end{vmatrix} \neq 0, \tag{27}$$

which is implied by $\lambda_k > 0$ for $1 \leq k \leq r, \lambda_i \neq \lambda_j, i \neq j$, assumed in Assumption 1. By Lemma 5, we have

$$\frac{1}{\prod_{j=0}^{a}(n-j)}\sum_{i=1}^{n}(L_i^{(a)} - \mathbb{E}(L_i^{(a)})) = O_p(n^{-1/2}) \text{ for } 1 \leq a \leq r,$$

$$\frac{1}{\prod_{j=0}^{a-1}(n-j)}\sum_{i=1}^{n}(C_i^{(a)} - \mathbb{E}(C_i^{(a)})) = O_p(n^{-1/2}) \text{ for } 3 \leq a \leq r+2.$$

Then by Lemma 6, we have

$$\max_{1 \leq k \leq r} |\hat{\lambda}_k - \lambda_k| = O_p(n^{-1/2}).$$

By Lemma 7, we have

$$\max_{1 \leq k \leq r} \left| y_k - \int_0^1 G_k(u)du \right| = O_p(n^{-1/2}).$$

We mention that there is no square root ambiguity since we assume $\int_0^1 G_i(u)du \geq 0, i = 1, 2$.

**(Step 2.)** In this step, we prove that $\sup_{i,j} |\hat{p}_{ij} - p_{ij}| = O_p(\sqrt{\log(n)/n})$. Recall that $(G_1(U_i), \cdots, G_r(U_i))$ is estimated by solving the system of equations with respect to $(\hat{G}_1(U_i), \cdots, \hat{G}_r(U_i))$:

$$\frac{1}{\prod_{j=1}^a (n-j)} L_i^{(a)} = \sum_{k=1}^r \hat{\lambda}_k^a y_k \hat{G}_k(U_i) \text{ for } 1 \leq a \leq r$$

with $\hat{\lambda}_k^a, y_k, 1 \leq k \leq r$ defined in (6). Note that for the above linear equation, we have

$$\max_i \max_k |\hat{G}_k(U_i) - G_k(U_i)| = O_p(\sqrt{\log(n)/n}) \tag{28}$$

as long as $\max_i \max_a |L_i^{(a)} - \mathbb{E}(L_i^{(a)}|U_i)| / \prod_{j=1}^a (n-j) = O_p(\sqrt{\log(n)/n})$, which is indeed indicated by Lemma 8.

According to (8), we have for every $1 \leq k \leq r$,

$$\tilde{G}_k(U_i) - \hat{G}_k(U_i) = \frac{\hat{G}_k(U_i)}{\sqrt{\sum_{i=1}^n \hat{G}_k^2(U_i)/n}} - \hat{G}_k(U_i) = \hat{G}_k(U_i) \left( \frac{1}{\sqrt{\sum_{i=1}^n \hat{G}_k^2(U_i)/n}} - 1 \right). \tag{29}$$

Since $U_i$'s are $i.i.d.$, there is $\sum_{i=1}^n G_k(U_i)^2/n - 1 = O_p(n^{-1/2})$. Hence we have

$$\sum_{i=1}^n \hat{G}_k(U_i)^2/n - 1 = \sum_{i=1}^n \hat{G}_k(U_i)^2/n - \sum_{i=1}^n G_k(U_i)^2/n + \sum_{i=1}^n G_k(U_i)^2/n - 1 = O_p(\sqrt{\log(n)/n}),$$

which implies that

$$\frac{1}{\sqrt{\sum_{i=1}^n \hat{G}_k^2(U_i)/n}} - 1 = O_p(\sqrt{\log(n)/n}). \tag{30}$$

By Assumption 1, $G_k$ are all bounded by $K$. Combining equation (30), (29), (28) and noting that $r = O(1)$, we have

$$\max_k \max_i |\tilde{G}_k(U_i) - \hat{G}_k(U_i)| = O_p(\sqrt{\log(n)/n}).$$

Therefore

$$\max_k \max_i |\tilde{G}_k(U_i) - G_k(U_i)| = O_p(\sqrt{\log(n)/n}).$$

As a result, in terms of the estimation of connection probabilities, we have

$$\sup_{i,j} |\hat{p}_{ij} - p_{ij}| = \sup_{i,j} |[1 \wedge (0 \vee (\sum_{k=1}^r \hat{\lambda}_k \tilde{G}_k(U_i) \tilde{G}_k(U_j))]$$

$$- (\sum_{k=1}^r \lambda_k G_k(U_i) G_k(U_j))| = O_p(\sqrt{\log(n)/n}).$$

$\square$

*Proof of Lemma 1.* We only show that

$$\max_{1 \leq i \leq n} \max_{1 \leq a \leq r} \left| \frac{1}{\prod_{j=1}^a (n-j)} (\tilde{L}_i^{(a)} - L_i^{(a)}) \right| = o_p \left( \frac{1}{\sqrt{n}} \right),$$

as the result for $\tilde{C}_i^{(a)}$ follows similarly.

By definition, we have

$$\tilde{L}_i^{(a)} - L_i^{(a)} = \sum_{i_1, \cdots, i_a \in \mathcal{M}} E_{i,i_1} \prod_{j=2}^a E_{i_{j-1}, i_j}$$

where $\mathcal{M} = \{\text{At least two of the values } i, i_1, \cdots, i_a \text{ are identical}\}$. Then

$$\frac{1}{\prod_{j=1}^{a}(n-j)}|\tilde{L}_i^{(a)} - L_i^{(a)}| \leq \frac{1}{\prod_{j=1}^{a}(n-j)} \sum_{i_1,\cdots,i_a \in \mathcal{M}} 1 = \frac{O(n^{a-1})}{\prod_{j=1}^{a}(n-j)}.$$

As a result,

$$\max_{1 \leq i \leq n} \max_{1 \leq a \leq r} \frac{1}{\prod_{j=1}^{a}(n-j)}|\tilde{L}_i^{(a)} - L_i^{(a)}| \leq \frac{O(n^{a-1})}{\prod_{j=1}^{a}(n-j)} = O_p\left(\frac{1}{n}\right).$$

$\square$

### A.6 TECHNICAL LEMMAS

**Lemma 2.** *For ASG(2) model with* $f(u,v) = \lambda_1 G_1(u)G_1(v) + \lambda_2 G_2(u)G_2(v)$ *with* $G_1, G_2$ *bounded by a constant* $M > 0$*, then we have*

$$\sup_{i=1,\cdots,n} \frac{|d_i - \mathbb{E}(d_i|U_i)|}{n-1} = O_p(\sqrt{\log(n)/n}),$$

*where* $d_i$ *is the degree of* $i_{th}$ *node. Note that the model reduces to ASG(1) when we set* $\lambda_2 = 0$.

*Proof.* We first note that

$$\sup_{i=1\cdots,n} \left| \frac{1}{n-1} \sum_{j:j\neq i} (\lambda_1 G_1(U_i)G_1(U_j) + \lambda_2 G_2(U_i)G_2(U_j)) \right.$$

$$\left. -\lambda_1 G_1(U_i)\int_0^1 G_1(u)du - \lambda_2 G_2(U_i)\int_0^1 G_2(u)du \right|$$

$$\leq \lambda_1 M \left( \left| \frac{1}{n-1}\sum_{j=1}^{n} G_1(U_j) - \int_0^1 G_1(u)du \right| + \frac{1}{n-1}M \right)$$

$$+ \lambda_2 M \left( \left| \frac{1}{n-1}\sum_{j=1}^{n} G_1(U_j) - \int_0^1 G_1(u)du \right| + \frac{1}{n-1}M \right) = O_p(n^{-1/2}),$$

where the last result follows from Slutsky's Theorem. Then it suffices to show that

$$\sup_{i=1\cdots,n} \left| \frac{1}{n-1} \sum_{j:j\neq i} (I(U_{ij} \leq \lambda_1 G_1(U_i)G_1(U_j) + \lambda_2 G_2(U_i)G_2(U_j)) \right. \tag{31}$$

$$\left. -\lambda_1 G_1(U_i)G_1(U_j) - \lambda_2 G_2(U_i)G_2(U_j)| = O_p(\sqrt{\log(n)/n}), \right. \tag{32}$$

where $U_{ij}, i \leq j$ are i.i.d. uniformly distributed random variables on $[0,1]$, and $U_{ji} = U_{ij}$ for $i > j$. Let

$$Z_i = \frac{1}{n-1} \sum_{j=1}^{n} (I(U_{ij} \leq \lambda_1 G_1(U_i)G_1(U_j) + \lambda_2 G_2(U_i)G_2(U_j))$$

$$-\lambda_1 G_1(U_i)G_1(U_j) - \lambda_2 G_2(U_i)G_2(U_j)).$$

By Hoeffding's inequality in Theorem 2.6.2 of Vershynin (2018), we have for any $t > 0$, $\mathbb{P}(\sqrt{n}|Z_i| > t|U_1, \cdots, U_n) \leq 2\exp(-ct^2)$ where $c > 0$ is an absolute constant. Then $\mathbb{P}(\sqrt{n}|Z_i| > t) = \mathbb{E}(\mathbb{P}(\sqrt{n}|Z_i| > t|U_1, \cdots, U_n)) \leq 2\exp(-ct^2)$. As a result, $\sqrt{n}Z_i$ are subgaussian random variables. Then we have $\mathbb{E}\max_{i=1\cdots,n}|Z_i| = O(\sqrt{\log(n)}/\sqrt{n})$, which indicates that $\max_{i=1\cdots,n}|Z_i| = O_p(\sqrt{\log(n)/n})$. $\square$

**Lemma 3.** *Suppose that* $U_i \overset{i.i.d.}{\sim} Uniform(0,1), i = 1, \cdots, n$. *Let* $U_{(i)}$ *denote the* $i$-th *smallest value among* $U_1, \cdots, U_n$, *i.e.,* $U_{(1)} \leq U_{(2)} \leq \cdots \leq U_{(n)}$. *Then*

$$\sup_i \left| U_{(i)} - \frac{i}{n+1} \right| \overset{a.s.}{\to} 0.$$

*Proof.* It is obvious that $U_{(i)} \sim Beta(i, n - i + 1)$ with a probability density function $p(x) = x^{i-1}(1-x)^{n-1}/\int_0^1 x^{i-1}(1-x)^{n-1}dx$. Then we derive that for any $\varepsilon > 0$, by Markov's inequality,

$$\mathbb{P}\left(\left|U_{(i)} - \frac{i}{n+1}\right| \geq \varepsilon\right) \leq \frac{1}{\varepsilon^6}\mathbb{E}\left|U_{(i)} - \frac{i}{n+1}\right|^6$$

$$= \frac{1}{\varepsilon^6}\frac{5i(n-i+1)A}{(n+1)^6(n+2)(n+3)(n+4)(n+5)(n+6)}$$

$$\leq \frac{1}{\varepsilon^6}\frac{5n^2A}{(n+1)^{11}}$$

where $A = 24(n-i+1)^4 + 2i(n-i+1)^3(13n-13i+1) + i^2(n-i+1)^2(24-8(n-i+1)+3(n-i+1)^2) + 2i^3(n-i+1)^2(3(n-i+1)^2 - 4(n-i+1) - 12) + i^4(24 + 26(n-i+1) + 3(n-i+1)^2)$. Note that $A \leq 12n^6 + 36n^5 + 24n^4 \leq 72n^6$. Then we have

$$\sum_{n=1}^{\infty}\mathbb{P}\left(\sup_i\left|U_{(i)} - \frac{i}{n+1}\right| \geq \varepsilon\right) \leq \sum_{n=1}^{\infty}\sum_{i=1}^{n}\mathbb{P}\left(\left|U_{(i)} - \frac{i}{n+1}\right| \geq \varepsilon\right)$$

$$\leq \frac{1}{\varepsilon^6}\sum_{n=1}^{\infty}\sum_{i=1}^{n}\frac{360n^8}{(n+1)^{11}}$$

$$\leq \frac{360}{\varepsilon^6}\sum_{n=1}^{\infty}\frac{1}{n^2} < \infty.$$

Therefore, by the Borel-Cantelli lemma, the result follows. $\qquad\square$

**Lemma 4.** *Let $G(u), u \in [0,1]$ be a monotonically non-decreasing, Lipschitz continuous function with Lipschitz constant $L > 0$. Let $a_i := G(i/(n+1)), i = 1, \cdots, n$. Suppose that there exists a sequence of random variables $b_1, \cdots, b_n$ such that $\sup_{i=1,\cdots,n}|b_i - a_i| \overset{a.s.}{\to} 0$. Let $\alpha$ be a one-to-one permutation such that $b_{\alpha(1)} \leq b_{\alpha(2)} \leq \cdots \leq b_{\alpha(n)}$. Let $\hat{a}_i := b_{\alpha(i)}$. Then we have $\sup_i|\hat{a}_i - a_i| \overset{a.s.}{\to} 0$. Moreover, if $\sup_{i=1,\cdots,n}|b_i - a_i| = O_p(g_n)$ then $\sup_i|\hat{a}_i - a_i| = O_p(g_n)$ for some $g_n = o(1), ng_n \to \infty$.*

*Proof.* Let $M_n = \sup_{i=1,\cdots,n}|b_i - a_i|$, then $M_n \overset{a.s.}{\to} 0$. Assume without loss of generality that $1/n = o_{a.s.}(M_n)$. Let $K_n$ be a non-negative random variable such that $K_n \overset{a.s.}{\to} 0, 3M_n \leq K_n \leq 4M_n, 1/n = o_{a.s.}(K_n)$. For any $i = 1, \cdots, n$, we have

$$|\hat{a}_i - a_i| = |b_{\alpha(i)} - a_i| \leq |a_{\alpha(i)} - a_i| + M_n.$$

First, consider the case where $\alpha(i) \geq i$. Assume, for the sake of contradiction, that $a_{\alpha(i)} - a_i > K_n$. Then for $j = 1, 2, \cdots, i+1$, we derive that

$$b_{\alpha(i)} \geq a_{\alpha(i)} - M_n > a_j - \frac{L}{n+1} + K_n - M_n \geq b_j - \frac{L}{n+1} + K_n - 2M_n,$$

where for the second inequality we use the monotonicity. By the construction of $K_n$, with probability 1, when $n$ is sufficiently large, we have $b_{\alpha(i)} > b_j, j = 1, 2, \cdots, i+1$. This implies that there are at least $i + 1$ values that are smaller than $b_{\alpha(i)}$, which contradicts the definition of $\alpha$. Therefore, $a_{\alpha(i)} - a_i \leq K_n$.

Similarly, for the case where $\alpha(i) \leq i$, we have that $a_{\alpha(i)} - a_i \geq -K_n$.

Then we conclude that

$$\sup_i|\hat{a}_i - a_i| = O_{a.s.}(K_n) + M_n \overset{a.s.}{\to} 0.$$

The statement of $O_p$ follows the exact same argument. $\qquad\square$

**Lemma 5.** *Under the assumptions of Theorem 3, we have*

$$\frac{1}{\prod_{j=0}^{a}(n-j)} \sum_{i=1}^{n} (L_i^{(a)} - \mathbb{E}(L_i^{(a)})) = O_p(n^{-1/2}) \, for \, 1 \le a \le r,$$

$$\frac{1}{\prod_{j=0}^{a-1}(n-j)} \sum_{i=1}^{n} (C_i^{(a)} - \mathbb{E}(C_i^{(a)})) = O_p(n^{-1/2}) \, for \, 3 \le a \le r+2,$$

*where $L_i^{(a)}, C_i^{(a)}$ are defined in Section 3.2.*

*Proof.* We only show that

$$\frac{1}{\prod_{j=0}^{a}(n-j)} \sum_{i=1}^{n} (L_i^{(a)} - \mathbb{E}(L_i^{(a)})) = O_p(n^{-1/2}) \text{ for } 1 \le a \le r,$$

as the results for $C_i^{(a)}$ follows similarly.

Note that $\mathbb{E}(E_{ij}|U_i, U_j) = f(U_i, U_j)$, and that $E_{ij}$ is conditional independent of $E_{i_1,j_1}$ when $(i,j) \ne (i_1, j_1)$. Then we derive that

$$\frac{1}{\left(\prod_{j=0}^{a}(n-j)\right)^2} \mathbb{E}\left[ \left( \sum_{i=1}^{n} L_i^{(a)} - \sum_{i=1}^{n} \mathbb{E}(L_i^{(a)}|U_1, \cdots, U_n) \right)^2 \middle| U_1, \cdots, U_n \right]$$

$$\lesssim \frac{1}{n^{2a+2}} \sum_{i,i_1,\cdots,i_a,k,k_1,\cdots,k_a} \mathbb{E}\left( \left( \left( E_{ii_1} \prod_{j=2}^{a} E_{i_{j-1}i_j} - f(U_i, U_{i_1}) \prod_{j=2}^{a} f(U_{i_{j-1}}, U_{i_j}) \right) \right. \right.$$

$$\left. \left. \left( E_{kk_1} \prod_{j=2}^{a} E_{k_{j-1}k_j} - f(U_k, U_{k_1}) \prod_{j=2}^{a} f(U_{k_{j-1}}, U_{k_j}) \right) \right) \middle| U_1, \cdots, U_n \right)$$

$$\lesssim \frac{n^{2a}}{n^{2a+2}} = \frac{1}{n^2}.$$

Since $\sum_{i=1}^{n} L_i^{(a)} \le \prod_{j=0}^{a}(n-j)$, we have

$$\frac{1}{\prod_{j=0}^{a}(n-j)} \sum_{i=1}^{n} (L_i^{(a)} - \mathbb{E}(L_i^{(a)}|U_1, \cdots, U_n)) = O_p\left(\frac{1}{n}\right). \tag{33}$$

Moreover, by the property of U-statistics (see for example, Theorem 4.2.1 in Korolyuk (2013)), we have

$$\frac{\sum_{i,i_1,\cdots,i_a} f(U_i, U_{i_1}) \prod_{j=2}^{a} f(U_{i_{j-1}}, U_{i_j})}{\prod_{j=0}^{a}(n-j)} = \frac{\mathbb{E}\sum_{i,i_1,\cdots,i_a} f(U_i, U_{i_1}) \prod_{j=2}^{a} f(U_{i_{j-1}}, U_{i_j})}{\prod_{j=0}^{a}(n-j)} + O_p(n^{-1/2}). \tag{34}$$

Note that

$$\sum_{i=1}^{n} \mathbb{E}(L_i^{(a)}|U_1, \cdots, U_n) = \sum_{i,i_1,\cdots,i_a} f(U_i, U_{i_1}) \prod_{j=2}^{a} f(U_{i_{j-1}}, U_{i_j}).$$

Then the result follows by combining (34) with (33). $\square$

**Lemma 6.** *Suppose that $x_1, \ldots, x_r$ are $r$ real numbers that satisfies $|x_1| > |x_2| > \cdots > |x_r| > 0$. Let $\epsilon_{3,n}, \ldots, \epsilon_{r+2,n}$ be $r$ random variables satisfying $\max_i |\epsilon_{i,n}| = O_p(n^{-1/2})$. Then the solution $(\tilde{x}_1, \ldots, \tilde{x}_r)$ for the following system of equations:*

$$\sum_{k=1}^{r} \tilde{x}_k^a = \sum_{k=1}^{r} x_k^a + \epsilon_{a,n} \, for \, 3 \le a \le r+2 \tag{35}$$

*satifies*

$$\max_i |\tilde{x}_i - x_i| = O_p(1/\sqrt{n}).$$

*Proof.* Let $\Delta_i = \tilde{x}_i - x_i, 1 \le i \le r$. By implicit function theorem, the system of equations (35) has one unique solution with probability tending to 1. Moreover, by the continuous mapping theorem, we have $\Delta_i = o_p(1)$. By the definition of $\epsilon_{i,n}$, for any $\varepsilon > 0$, there exists a finite $M$ and a finite $N$ such that

$$\mathbb{P}(\max_i |\sqrt{n}\epsilon_i| > M) < \varepsilon, \forall n > N.$$

Therefore, it suffices to show that

$$\mathbb{P}(\max_i |\Delta_i| \le C \max_i |\epsilon_{i,n}|) \to 1 \tag{36}$$

for some constant $C > 0$. Note that

$$\sum_{k=1}^r \tilde{x}_k^a - \sum_{k=1}^r x_k^a = \sum_{k=1}^r (x_k + \Delta_k)^a - \sum_{k=1}^r x_k^a = \sum_{k=1}^r a x_k^{a-1} \Delta_k + O_p(\max_k \Delta_k^2).$$

We then calculate that

$$\sum_{k=1}^r a x_k^{a-1} \Delta_k = \tilde{\epsilon}_{a,n} \text{ for } 3 \le a \le r+2$$

where $\tilde{\epsilon}_{a,n} = \delta_a + \epsilon_{a,n}, \delta_a = O_p(\max_i \Delta_i^2)$. For the above linear system of equations, by our assumption on $x_i$ (similar to the arguments in (27)), it has one unique solution with the form

$$\Delta_i = \sum_{j=3}^{r+2} a_{i,j} \tilde{\epsilon}_{j,n} \tag{37}$$

where $a_{i,j}$ are constants depend on $x_1, \ldots, x_r$ only. By combining (37) and the fact that $\max_a |\delta_a| = O_p(\max_i \Delta_i^2), \Delta_i = o_p(1)$, we conclude that (36) follows. $\square$

**Lemma 7.** *Suppose that $x_1, \ldots, x_r$ are $r$ real numbers that satisfies $|x_1| > |x_2| > \cdots > |x_r| > 0$, $\tilde{x}_1, \ldots, \tilde{x}_r$ are $r$ random variables that satisfies $\max_i |\tilde{x}_i - x_i| = O_p(1/\sqrt{n})$. Let $y_1, \ldots, y_r$ be $r$ non-zero real numbers, $\epsilon_{1,n}, \ldots, \epsilon_{r,n}$ be $r$ random variables satisfying $\max_i |\epsilon_{i,n}| = O_p(n^{-1/2})$. Then the solution $(\tilde{y}_1, \cdots, \tilde{y}_r)$ for the following system of equations with respect to $(y_1, \cdots, y_r)$:*

$$y_a \ge 0, \sum_{k=1}^r \tilde{x}_k^a \tilde{y}_k^2 = \sum_{k=1}^r x_k^a y_k^2 + \epsilon_{a,n} \text{ for } 1 \le a \le r \tag{38}$$

*satifies*

$$\max_i |\tilde{y}_i - y_i| = O_p(1/\sqrt{n}).$$

*Proof.* Note that

$$\tilde{x}_k^a \tilde{y}_k^2 - x_k^a y_k^2 = (\tilde{x}_k^a - x_k^a) y_k^2 + \tilde{x}_k^a (\tilde{y}_k^2 - y_k^2).$$

Since $\max_i |\tilde{x}_i - x_i| = O_p(1/\sqrt{n})$, we have $\max_i |\tilde{x}_i^a - x_i^a| = O_p(1/\sqrt{n})$. Then (38) reduces to

$$y_a \ge 0, \sum_{k=1}^r \tilde{x}_k^a (\tilde{y}_k^2 - y_k^2) = \tilde{\epsilon}_{a,n} \text{ for } 1 \le a \le r$$

where $\max_a |\tilde{\epsilon}_{a,n}| = O_p(n^{-1/2})$. Moreover, since $\max_k |\tilde{x}_k^a| = O_p(1)$, by noticing that the above system of equations is a linear system with respect to $\tilde{y}_k^2 - y_k^2, 1 \le k \le r$, and that $r = O(1)$, we have $\max_k |\tilde{y}_k^2 - y_k^2| = O_p(n^{-1/2})$. Finally, recalling that $y_1, \cdots, y_r$ are non-zero, we have $\max_k |\tilde{y}_k - y_k| = O_p(n^{-1/2})$. $\square$

**Lemma 8.** *Under the assumptions of Theorem 3, we have*

$$\max_{1 \le i \le n} \max_{1 \le a \le r} |L_i^{(a)} - \mathbb{E}(L_i^{(a)}|U_i)| / \prod_{j=1}^a (n-j) = O_p(\sqrt{\log(n)/n})$$

*where*

$$L_i^{(1)} = \sum_{i_1} E_{ii_1}, \quad L_i^{(a)} = \sum_{i_1, \cdots, i_a \text{ distinct}, i_k \ne i, 1 \le k \le a} E_{ii_1} \prod_{j=2}^a E_{i_{j-1}i_j} \text{ for } a \ge 2.$$

*Proof.* We divide the proof into two steps. In **Step 1**, we show that

$$\frac{1}{\prod_{j=1}^a (n-j)} \max_i |L_i^{(a)} - S_{i,0}| = O_p(\sqrt{\log(n)/n})$$

where

$$S_{i,0} = \sum_{i_1,\cdots,i_a \text{ distinct },i_k \neq i, 1 \leq k \leq a} f(U_i, U_{i_1}) \prod_{j=2}^a f(U_{i_{j-1}}, U_{i_j}).$$

In **Step 2**, we show that

$$\frac{1}{\prod_{j=1}^a (n-j)} \max_i |S_{i,0} - T_{i,1}| = O_p(n^{-1/2})$$

where

$$T_{i,1} = \mathbb{E}\left[ \sum_{i_1,\cdots,i_a \text{ distinct },i_k \neq i, 1 \leq k \leq a} f(U_i, U_{i_1}) \prod_{j=2}^a f(U_{i_{j-1}}, U_{i_j}) \Big| U_i \right] = \mathbb{E}(L_i^{(a)}|U_i).$$

Then the proof is complete by combining the above two equations and noticing that $r$ is bounded.

**Step 1.** Let

$$S_{i,a-1} = \sum_{i_1,\cdots,i_a \text{ distinct },i_k \neq i, 1 \leq k \leq a} E_{ii_1} \prod_{j=2}^{a-1} E_{i_{j-1}i_j} f(U_{i_{a-1}}, U_{i_a}).$$

Then

$$\frac{1}{\prod_{j=1}^a (n-j)} (L_i^{(a)} - S_{i,a-1}) = \frac{1}{\prod_{j=1}^a (n-j)} \sum_{i_1,\cdots,i_a \text{ distinct },i_k \neq i, 1 \leq k \leq a} E_{ii_1} \prod_{j=2}^{a-1} E_{i_{j-1}i_j} (E_{i_{a-1}i_a} - f(U_{i_{a-1}}, U_{i_a})).$$

$$(39)$$

Notice that $E_{i_{a-1}i_a} = I(U_{i_{a-1},i_a} \leq f(U_{i_{a-1}}, U_{i_a}))$ is binary, $U_{i_{a-1},i_a} \sim Uniform(0,1)$ independently, and that $U_{ij}$ is independent of $U_k$ for any $i, j, k$. By Hoeffding's inequality in Theorem 2.6.2 of Vershynin (2018), we have for any $t > 0$,

$$\mathbb{P}\left( \frac{1}{\sqrt{n-a}} \left| \sum_{i_a \neq i, i_1,\cdots,i_{a-1}} (E_{i_{a-1}i_a} - f(U_{i_{a-1}}, U_{i_a})) \right| \geq t \Big| U_1,\cdots,U_n \right) \leq 2\exp(-ct^2)$$

where $c > 0$ is an absolute constant. Then

$$\mathbb{P}\left( \frac{1}{\sqrt{n-a}} \left| \sum_{i_a \neq i, i_1,\cdots,i_{a-1}} (E_{i_{a-1}i_a} - f(U_{i_{a-1}}, U_{i_a})) \right| \geq t \right)$$

$$= \mathbb{E}\left( \mathbb{P}\left( \frac{1}{\sqrt{n-a}} \left| \sum_{i_a \neq i, i_1,\cdots,i_{a-1}} (E_{i_{a-1}i_a} - f(U_{i_{a-1}}, U_{i_a})) \right| \geq t \Big| U_1,\cdots,U_n \right) \right) \leq 2\exp(-ct^2).$$

As a result, $\frac{1}{\sqrt{n-a}} \left| \sum_{i_a \neq i, i_1,\cdots,i_{a-1}} (E_{i_{a-1}i_a} - f(U_{i_{a-1}}, U_{i_a})) \right|$ are sub-gaussian random variables, and we have $\mathbb{E}\max_{i_{a-1}} \left| \sum_{i_a \neq i, i_1,\cdots,i_{a-1}} E_{i_{a-1}i_a} - f(U_{i_{a-1}}, U_{i_a}) \right| / (n-a) = O(\sqrt{\log(n)/n})$. By recalling (39) and the fact that $E_{ij}$'s are binary, we have

$$\frac{1}{\prod_{j=1}^a (n-j)} \mathbb{E}\max_i |L_i^{(a)} - S_{i,a-1}| = O(\sqrt{\log(n)/n}).$$

Similarly, let

$$S_{i,a-2} = \sum_{i_1,\cdots,i_a \text{ distinct },i_k \neq i, 1 \leq k \leq a} E_{ii_1} \prod_{j=2}^{a-2} E_{i_{j-1}i_j} f(U_{i_{a-2}}, U_{i_{a-1}}) f(U_{i_{a-1}}, U_{i_a}).$$

Then

$$\frac{1}{\prod_{j=1}^{a}(n-j)}(S_{i,a-1} - S_{i,a-2}) = \frac{1}{\prod_{j=1}^{a}(n-j)} \sum_{i_1,\cdots,i_a \text{ distinct},i_k \neq i,1\leq k\leq a} E_{ii_1} \times$$

$$\prod_{j=2}^{a-2} E_{i_{j-1}i_j}(E_{i_{a-2}i_{a-1}} - f(U_{i_{a-2}}, U_{i_{a-1}}))f(U_{i_{a-1}}, U_{i_a})$$

$$= \frac{1}{\prod_{j=1}^{a-1}(n-j)} \sum_{i_1,\cdots,i_{a-1} \text{ distinct},i_k \neq i,1\leq k\leq a-1} E_{ii_1} \times$$

$$\prod_{j=2}^{a-2} E_{i_{j-1}i_j}(E_{i_{a-2}i_{a-1}} - f(U_{i_{a-2}}, U_{i_{a-1}}))\frac{1}{n-a}\sum_{i_a} f(U_{i_{a-1}}, U_{i_a}).$$

$$(40)$$

By Hoeffding's inequality and noticing that the terms $(E_{i_{a-2}i_{a-1}} - f(U_{i_{a-2}}, U_{i_{a-1}}))\frac{1}{n-a}\sum_{i_a} f(U_{i_{a-1}}, U_{i_a})$ are bounded, we have for any $t > 0$,

$$\mathbb{P}\left(\frac{1}{\sqrt{n-a+1}}\left|\sum_{i_{a-1}}(E_{i_{a-2}i_{a-1}} - f(U_{i_{a-2}}, U_{i_{a-1}}))\frac{1}{n-a}\sum_{i_a} f(U_{i_{a-1}}, U_{i_a})\right| \geq t \middle| U_1,\cdots,U_n\right) \leq 2\exp(-c't^2)$$

where $c' > 0$ is an absolute constant. Then

$$\frac{1}{\sqrt{n-a+1}}\left|\sum_{i_{a-1}:i_{a-1}\neq i,i_1,\cdots,i_{a-2}}(E_{i_{a-2}i_{a-1}} - f(U_{i_{a-2}}, U_{i_{a-1}}))\frac{1}{n-a}\sum_{i_a:i_{a-1}\neq i,i_1,\cdots,i_{a-1}} f(U_{i_{a-1}}, U_{i_a})\right|$$

for any $i, i_1, \cdots, i_{a-2}$, are sub-gaussian random variables, and

$$\mathbb{E}\max_{i,i_1,\cdots,i_{a-2}} \frac{1}{\sqrt{n-a+1}}\left|\sum_{i_{a-1}:i_{a-1}\neq i,i_1,\cdots,i_{a-2}}(E_{i_{a-2}i_{a-1}} - f(U_{i_{a-2}}, U_{i_{a-1}}))\times\right.$$

$$\left.\frac{1}{n-a}\sum_{i_a:i_{a-1}\neq i,i_1,\cdots,i_{a-1}} f(U_{i_{a-1}}, U_{i_a})\right| = O(\sqrt{\log(n)}).$$

By recalling (40) and the fact that $E_{ij}$'s are binary,, $a = O(1)$, we have

$$\frac{1}{\prod_{j=1}^{a}(n-j)}\mathbb{E}\max_i |S_{i,a-1} - S_{i,a-2}| = O(\sqrt{\log(n)/n}).$$

Similar arguments can be perfomed for $S_{i,a-3},\cdots,S_{i,1},S_{i,0}$ (we define $i_0 = i$). Since $a \leq r$ is bounded, by combining all the results, we have

$$\frac{1}{\prod_{j=1}^{a}(n-j)}\mathbb{E}\max_i |L_i^{(a)} - S_{i,0}| = O(\sqrt{\log(n)/n}) \tag{41}$$

where

$$S_{i,0} = \sum_{i_1,\cdots,i_a \text{ distinct},i_k \neq i,1\leq k\leq a} f(U_i, U_{i_1})\prod_{j=2}^{a} f(U_{i_{j-1}}, U_{i_j}).$$

Then

$$\frac{1}{\prod_{j=1}^{a}(n-j)}\max_i |L_i^{(a)} - S_{i,0}| = O_p(\sqrt{\log(n)/n}).$$

**Step 2.** Let

$$T_{i,a-1} = \sum_{i_1,\cdots,i_a \text{ distinct},i_k \neq i,1\leq k\leq a} f(U_i, U_{i_1})\prod_{j=2}^{a-1} f(U_{i_{j-1}}, U_{i_j})\mathbb{E}(f(U_{i_{a-1}}, U_{i_a})|U_{i_{a-1}}).$$

Then

$$\max_i \frac{1}{\prod_{j=1}^a (n-j)} |S_{i,0} - T_{i,a-1}|$$

$$= \max_i \frac{1}{\prod_{j=1}^a (n-j)} \left| \sum_{i_1,\cdots,i_a \text{ distinct },i_k \neq i, 1 \leq k \leq a} f(U_i, U_{i_1}) \prod_{j=2}^{a-1} f(U_{i_{j-1}}, U_{i_j})(f(U_{i_{a-1}}, U_{i_a}) - \mathbb{E}(f(U_{i_{a-1}}, U_{i_a})|U_{i_{a-1}})) \right|$$

$$= \max_i \frac{1}{\prod_{j=1}^{a-1} (n-j)} \left| \sum_{i_1,\cdots,i_{a-1} \text{ distinct },i_k \neq i, 1 \leq k \leq a-1} f(U_i, U_{i_1}) \prod_{j=2}^{a-1} f(U_{i_{j-1}}, U_{i_j}) \right.$$

$$\left. \frac{1}{n-a} \sum_{i_a} \sum_{k=1}^r \lambda_k G_k(U_{i_{a-1}})[G_k(U_{i_a}) - \int_0^1 G_k(u)du] \right|$$

$$\leq \frac{1}{n-a} \sum_{k=1}^r \left| \lambda_k M \sum_{i_a} [G_k(U_{i_a}) - \int_0^1 G_k(u)du] \right| = O_p(n^{-1/2}),$$

where we use the fact that $f(x,y)$ are bounded by 1, $G_k$ are bounded by $M$, and that $U_i$ are i.i.d. random variables.

Similarly, let

$$T_{i,a-2} = \sum_{i_1,\cdots,i_a \text{ distinct },i_k \neq i, 1 \leq k \leq a} f(U_i, U_{i_1}) \prod_{j=2}^{a-2} f(U_{i_{j-1}}, U_{i_j}) \mathbb{E}(f(U_{i_{a-2}}, U_{i_{a-1}})f(U_{i_{a-1}}, U_{i_a})|U_{i_{a-2}}).$$

Then

$$\max_i \frac{1}{\prod_{j=1}^a (n-j)} |T_{i,a-1} - T_{i,a-2}|$$

$$= \max_i \frac{1}{\prod_{j=1}^a (n-j)} \left| \sum_{i_1,\cdots,i_a \text{ distinct },i_k \neq i, 1 \leq k \leq a} f(U_i, U_{i_1}) \prod_{j=2}^{a-2} f(U_{i_{j-1}}, U_{i_j}) \right.$$

$$\left. (f(U_{i_{a-2}}, U_{i_{a-1}})\mathbb{E}(f(U_{i_{a-1}}, U_{i_a})|U_{i_{a-1}}) - \mathbb{E}(f(U_{i_{a-2}}, U_{i_{a-1}})f(U_{i_{a-1}}, U_{i_a})|U_{i_{a-2}})) \right|$$

$$\lesssim \frac{1}{n} \left| \sum_{i_{a-1}} \sum_{k_1=1}^r \sum_{k_2=1}^r \lambda_{k_1} \lambda_{k_2} G_{k_1}(U_{i_{a-1}}) G_{k_2}(U_{i_{a-1}}) \int_0^1 G_{k_2}(u)du - \sum_{i_{a-1}} \sum_{k=1}^r \lambda_k^2 \int_0^1 G_k(u)du \right|$$

$$\lesssim \frac{1}{n} \sum_{k=1}^r \left| \sum_{i_{a-1}} (G_k^2(U_{i_{a-1}}) - 1) \right| + \frac{1}{n} \sum_{k_1 \neq k_2} \left| \sum_{i_{a-1}} G_{k_1}(U_{i_{a-1}}) G_{k_2}(U_{i_{a-1}}) \right| + O\left(\frac{1}{n}\right)$$

$$= O_p(n^{-1/2}),$$

where we use the fact that $f(x,y)$ are bounded by 1, $G_k$ are bounded by $M$, $r$ is bounded, $\int_0^1 G_k^2(u)du = 1$, $\int_0^1 G_i(u)G_j(u)du = 0$ for $i \neq j$, and that $U_i$ are i.i.d. random variables.

Similar arguments can be perfomed for $T_{i,a-3}, \cdots, T_{i,1}$. Since $a \leq r$ is bounded, by combining all the results, we have

$$\frac{1}{\prod_{j=1}^a (n-j)} \max_i |S_{i,0} - T_{i,1}| = O_p(n^{-1/2}) \tag{42}$$

where

$$T_{i,1} = \mathbb{E}\left[ \sum_{i_1,\cdots,i_a \text{ distinct },i_k \neq i, 1 \leq k \leq a} f(U_i, U_{i_1}) \prod_{j=2}^a f(U_{i_{j-1}}, U_{i_j}) \middle| U_i \right].$$

Then the proof is complete by combining the results from (41), (42), and noticing that $r$ is bounded.

$\square$

