# OpenReview forum: "ADDITIVE SEPARABLE GRAPHON MODELS"
_ICLR.cc/2025/Conference — Submitted to ICLR 2025_

### Official Review · Reviewer_whmb · 2024-11-01

**Soundness:** 2
**Presentation:** 3
**Contribution:** 2
**Rating:** 6
**Confidence:** 4

**Summary:**

This paper proposes an additive separable model for graphons, producing a low-rank connection matrix and addressing certain identification issues. An estimation method using subgraph counts has been proposed to estimate the graph parameters. Several numerical experiments are showed to highlight the performance.

**Strengths:**

1. The proposed generalization approach is simple and easy to follow.
2. The bounds and minmax rates obtained are reasonable and the authors did a good job in explaining the results and the implications.
2. The numerical experiments encompasses a wide class of gryphon functions and several competing methods. Unlike cherry picking the results, the authors also showed scenarios where their method did not perform well.

**Weaknesses:**

The utility of the proposed generalization is questionable from both theoretical and empirical perspectives. Empirically, as shown in the comparison tables, the proposed method yields only marginal improvement over existing methods. Theoretically, it raises the question: why is an additive separable scheme considered reasonable? Why not a multiplicative or another type of schema?

**Questions:**

n/a

---

> ### Author Response · Authors · 2024-11-18
> **Response letter to Reviewer whmb (part 1)**
>
> Thank you for your interest in our paper and for your thoughtful review. We appreciate your positive feedback on the clarity of our method, as well as our theoretical results and simulation results. We summarize the main changes according to the valuable suggestions from reviewers as follows.
>
> 1.  We provide a complete theoretical framework for general $r$, updated in Lines 203–338 on pages 4–7.
> 2. Computationally, we introduce a new approximation approach for computations, updated in Lines 339–381 on pages 7–8. None of the theoretical results are affected, while it achieves a time complexity matching that of matrix multiplication ($ O(n^{2.373})$), eliminating the need for subsampling techniques.
> 3. In Appendix A.3, we provide a method for selecting $r$ when it is unknown.  We also point out this in Remark 2 of the main article.
> 4. We update the simulation results and included an additional simulation for $ r = 3 $.
> 5. In Appendix A.1, we include a real data analysis, demonstrating the use of our method for selecting $r$, as well as our algorithms for estimating the connection matrix and the graphon function.
> 6. We update the mathematical proofs for all theoretical results.
>
> In response to your questions, we have made revisions and improvements to the paper. Below, we address your points one by one.
>
> - *``Empirically, as shown in the comparison tables, the proposed method yields only marginal improvement over existing methods.''*
>
> Answer: We would like to clarify that the improvement in performance is influenced by the sparsity of the graphon. For relatively dense graphons (see Table 2), our method performs similarly to USVT, as both achieve minimax optimal rates for estimating the connection probability matrix. However, for sparse graphons (see Table 3), our method significantly outperforms USVT, achieving MSE reductions of over 50\%. This underscores the versatility of our approach in effectively addressing both dense and sparse networks.
>
> Importantly, this superior performance is accompanied by significantly lower computational requirements, with a complexity of $O(n^{2.373})$ (see Table 2). A further advantage of our method is that it is tuning-parameter-free, unlike the SAS method (Chan and Airoldi, 2014) and the network histogram method (Olhede and Wolfe, 2014), both of which require careful selection of tuning parameters. Consequently, these methods can be sensitive to the choice of parameters, potentially impacting their performance (see tables below).
>
> Additionally, our method directly leverages the low-rank structure of the graphon to estimate the graphon function itself, not just on discrete grids (as represented by connection probability matrices), but across the entire domain of $[0,1] \times [0,1]$. In contrast, neither the USVT method nor the neighborhood smoothing method can estimate the full graphon function.
>
> Thus, our method offers a computationally efficient (tuning-free with optimal complexity), theoretically optimal (achieving minimax rates), and comprehensive framework for graphon estimation, outperforming its competitors in these key aspects.
>
> ### The MSE of the SAS (Chan and Airoldi, 2014) under different parameter selections.
> | K   | 190   | 210   | 230   | 250 (default)   | 270   | 290   | 310   |
> |-----|-------|-------|-------|-------|-------|-------|-------|
> | MSE | 0.00155 | 0.00192 | 0.00235 | 0.00283 | 0.00340 | 0.00398 | 0.00462 |
>
> ### The MSE of the Nethist (Olhede and Wolfe, 2014) under different parameter selections.
> | h     | 30    | 40    | 50 (default)   | 60    | 70      |
> |------|-------|-------|-------|-------|------- |
> | MSE |  0.000873 | 0.000713 | 0.000617 | 0.000582 | 0.000525  |

---

> > ### Comment · Reviewer_whmb · 2024-11-27
> > **Thanks for your comments. I've updated my score**
> >
> > Based on the authors' responses to both my feedback and that of other reviewers, I have increased my score.

---

> ### Author Response · Authors · 2024-11-18
> **Response letter to Reviewer whmb (part 2)**
>
> - *``Theoretically, it raises the question: why is an additive separable scheme considered reasonable? Why not a multiplicative or another type of schema?''*
>
> Answer: Thank you for your question. The additive separable form is a natural choice, grounded in the insight provided by the Hilbert–Schmidt theorem (see for example [1]), which implies that any bounded graphon function $ f(u,v) $ can be decomposed as:  $f(u,v) = \sum_{j=1}^\infty \lambda_j G_j(u) G_j(v),$ where $G_j$ are orthonormal eigenfunctions and $\sum_{j} \lambda_j^2 < \infty$ [2].
>
> A practical approach to estimating this infinite series is to truncate it at $r$, retaining the $r$ eigenfunctions corresponding to  the $r$ largest absolute eigenvalues. This results in a low-rank structure that aligns naturally with our model and facilitates efficient estimation. An additional advantage of this approach is that the resulting connection probability matrix $P$—formed by evaluating the graphon function at the nodes—naturally inherits the same low-rank property as the truncated graphon.  We believe this model provides a robust and insightful framework for investigation. That said, exploring multiplicative or other alternative schemas remains an important direction for future research.
>
> [1] Szegedy, B. (2011). Limits of kernel operators and the spectral regularity lemma. European Journal of Combinatorics, 32(7), 1156–1167.
>
> [2] Bickel, P. J., Chen, A., & Levina, E. (2011). The method of moments and degree distributions for network models. The Annals of Statistics, 39(5), 2280–2301.

---

> ### Author Response · Authors · 2024-11-26
> **Main changes in the second revision**
>
> We have made a second revision to our article, and the main changes are listed below for your reference. In the newly revised version, we have revised the wording to reduce the emphasis on the novelty of the model itself. Instead, we frame our estimation approach as a new attempt to estimate a low-rank representation. Additionally, we have added a new remark (Remark 1) that discusses the power iteration method and included it in our simulation studies (see Tables 2, 3, and 5). Furthermore, we have added Remark 3 (line 289) in the revised manuscript to further highlight the key differences between our method and spectral methods, including differences in motivation, estimation procedure, assumptions, and empirical performance for sparse graphons. It is worth noting that our approach allows for the simultaneous estimation of both the connection probability matrix and the graphon function. For estimating the connection probability matrix, our method shows clear advantages over spectral methods in the sparse region (see Table 3).

---

### Official Review · Reviewer_LuY1 · 2024-11-01

**Soundness:** 3
**Presentation:** 3
**Contribution:** 2
**Rating:** 6
**Confidence:** 3

**Summary:**

This paper explores additive separable graphon models (ASG) as a a flexible framework to capture low-rank structures in network data. The authors propose simple and efficient algorithm based on subgraph counts for probability connection matrix rank $r$ either being $1$ or $2$, and further use interpolation to recover the graphon functions. A wide range of simulations are included to back up the performance of their method.

**Strengths:**

- The general framework of additive separable graphon models is appealing, which naturally introduces the low-rankness for network data.
- I do enjoy reading the paper, as it presents ideas in a clear and logical way. This makes the methodology and its contributions straightforward to follow.
- The simulations showcase both the efficiency and effectiveness of their method. This makes it easy to assess how the method performs across a range of synthetic network settings.

**Weaknesses:**

-  Overall, I would regard the proposed method as a  method for  estimating the network moment or motifs. That said, I doubt that it would scale easily to even moderate values of $r$, especially due to computational demands. This constraint could make it less practical for most commonly studied stochastic block models, where the connection probability matrix rank is often higher than $2$.

- Despite what’s suggested in the discussion, I think it would help if the authors could explicitly address the estimation process for general ASG(r) as this is particularly important for practitioners looking to apply it more broadly.

- As a follow-up question, deciding the appropriate rank $r$ could be a challenge in practice. A brief discussion on rank selection—whether there’s a heuristic, or a data-driven way to guide practitioners would make the approach more usable.

- Given the wide range of applications inspired by network literature, I suggest the authors analyze at least one real-world dataset to demonstrate the practical utility of their method.

**Questions:**

See weakness part.

---

> ### Author Response · Authors · 2024-11-18
> **Response letter to Reviewer LuY1**
>
> Thank you for your interest in our paper and for your thoughtful review. We appreciate your acknowledgment of the logical structure of our writing and the positive feedback on the simulation results. We summarize the main changes according to the valuable suggestions from reviewers as follows.
>
> 1.  We provide a complete theoretical framework for general $r$, updated in Lines 203–338 on pages 4–7.
> 2. Computationally, we introduce a new approximation approach for computations, updated in Lines 339–381 on pages 7–8. None of the theoretical results are affected, while it achieves a time complexity matching that of matrix multiplication ($O(n^{2.373})$), eliminating the need for subsampling techniques.
> 3. In Appendix A.3, we provide a method for selecting $r$ when it is unknown. We also point out this in Remark 2 of the main article.
> 4. We update the simulation   results and included an additional simulation for $r = 3 $.
> 5. In Appendix A.1, we include a real data analysis, demonstrating the use of our method for selecting $r$, as well as our algorithms for estimating the connection matrix and the graphon function.
> 6. We update the mathematical proofs for all theoretical results.
>
> In response to your questions, we have made revisions and enhancements to the manuscript. Below, we address your questions point by point.
>
> - *``Overall, I would regard the proposed method as a method for estimating the network moment or motifs. That said, I doubt that it would scale easily to even moderate values of $r$, especially due to computational demands. This constraint could make it less practical for most commonly studied stochastic block models, where the connection probability matrix rank is often higher than 2.''*
>
> Answer: Thank you for your suggestion, which motivated us to update the theory and implementation for the general case of $r$. In the revised version of our paper, we have included a comprehensive theoretical study for the case $r > 2$, which is now detailed in Lines 203–338 on page 4-7 of the revised version of the paper.  In the more general setting, we obtain the same consistent result as before. Computationally,  we find a new approximation approach for computing and update it in Lines 339-381 on pages 7-8. Specifically, we approximate simple paths with paths allowing node repetition and prove their asymptotic equivalence, ensuring that this approximation does not affect any theoretical results. This approach enables our method to achieve a time complexity matching that of matrix multiplication ($ O(n^{2.373}) $), eliminating the need for subsampling techniques.  We update all the simulation results using the new methods, and consider a new simulation scenario with rank $r=3$, see ID 7 in Tables 2. Additionally, we consider the problem of selecting $r$ in practice, see discussion on Algorithm 4 in Appendix A.3. and Table 6 for the corresponding simulation results.
>
> - *``Despite what’s suggested in the discussion, I think it would help if the authors could explicitly address the estimation process for general ASG(r) as this is particularly important for practitioners looking to apply it more broadly.''*
>
> Answer: Similar to the previous answer, we have now incorporated the theory  as well as the methods for choosing $r$ in practice. Thank you for your suggestion.
>
> - *``As a follow-up question, deciding the appropriate rank $r$ could be a challenge in practice. A brief discussion on rank selection—whether there’s a heuristic, or a data-driven way to guide practitioners would make the approach more usable.''*
>
> Answer: Thanks a lot! We have proposed a  computational feasible methods to sequentially determine $r$, which works well in our numerical analysis, see discussion on Algorithm 4 in Appendix A.3. and Table 6 for the corresponding simulation results.
>
> - *``Given the wide range of applications inspired by network literature, I suggest the authors analyze at least one real-world dataset to demonstrate the practical utility of their method.''*
>
> Answer: Thank you for your suggestion. We have added a real data example in section A.1 of the appendix (page 12-13), which comes from contact records in a primary school. Using this data, we selected $ r = 4 $ and estimated the corresponding connection matrix and graphon function.

---

> > ### Comment · Reviewer_LuY1 · 2024-11-24
> >
> > I appreciate the effort the authors have made in addressing my concerns, and I have updated my score in light of these improvements.

---

> > > ### Author Response · Authors · 2024-11-27
> > >
> > > Thank you very much for your response and your positive feedback on our revisions. We truly appreciate it.
> > >
> > > Also, we would like to let you know our second revision. The main changes are listed below. In the newly revised version, we have revised the wording to reduce the emphasis on the novelty of the model itself. Instead, we frame our estimation approach as a new attempt to estimate a low-rank representation. Additionally, we have added a new remark (Remark 1) that discusses the power iteration method and included it in our simulation studies (see Tables 2, 3, and 5). Furthermore, we have added Remark 3 (line 289) in the revised manuscript to further highlight the key differences between our method and spectral methods, including differences in motivation, estimation procedure, assumptions, and empirical performance for sparse graphons. It is worth noting that our approach allows for the simultaneous estimation of both the connection probability matrix and the graphon function. For estimating the connection probability matrix, our method shows clear advantages over spectral methods in the sparse region (see Table 3).
> > >
> > > Once again, we sincerely appreciate your thoughtful review and valuable suggestions for our article.

---

### Official Review · Reviewer_3djw · 2024-11-02

**Soundness:** 3
**Presentation:** 3
**Contribution:** 2
**Rating:** 6
**Confidence:** 4

**Summary:**

This paper proposes a method for estimating both the connection probability matrix and the graphon function for low-rank graphons of rank 1 and 2. The authors provide finite-sample error bounds for the connection probability matrix estimation, specifically in the max-norm (i.e., $\|A\|_ {max} = \max_ {ij} |A_{ij}|$). For the graphon function estimation, they establish both asymptotic and finite-sample bounds under the sup-norm on $[0,1]^2$. The experimental results on synthetic data highlight the method's performance across various rank-1 and rank-2 graphons.

**Strengths:**

1- **Clarity and Mathematical Rigor**: The paper is generally well-written and accessible, with clear explanations that guide the reader through the methodology.  The methods are simple, and the proofs appear mathematically sound and contribute to a solid theoretical foundation.

2- **Competitive Experimental Results**: The proposed methods demonstrate competitive performance on synthetic datasets.

**Weaknesses:**

1- **Novelty**: Regarding the proposed model Additive Separable Graphons ASG($r$), the authors state in line 076 that this model is new. However, it appears to me as a generic low-rank graphon, which is a well-recognized particular case of a general graphon already discussed in the literature (e.g., Chan and Airoldi, 2014; Xu, 2018). Indeed, for any rank-$r$ graphon, the decomposition given in Eq. $(1)$ follows directly from the spectral theorem. I struggle to see this as a new model, and in my opinion, it would be beneficial for the authors to justify why ASG($r$) is indeed novel.

In terms of the methods, **Algorithm 1** resembles a step of the power iteration applied to the adjacency matrix $E$ with the starting vector $\mathbf{1}$ (the vector of all ones).  Taking $v= \frac{E\mathbf{1}}{\|E\mathbf{1}\|_ 2}$, we have $v_i=\frac{\sum_ {j}E_{ij}}{\|E\mathbf{1}\|_ 2}=\frac{d_i}{\|d_i\|_ 2}$, where $d_i$ is the degree of the node $i$, as in the paper. On the other hand, the estimation of the eigenvalue is given by the Rayleigh quotient $\lambda'_ 1=\frac{\mathbf{1}^\top E \mathbf{1}}{n}=\frac{\sum_ {i,j}E_{ij}}{n}=\frac{\sum_ {i,j:i\neq j}E_{ij}}{n}$. With this normalization $v_i$ is an estimation of $\frac{G_1(U_i)}{\sqrt{n}}$, then putting this together yields an estimation of probability matrix: $p'_ {ij}=n\lambda'_ 1(vv^\top)_ {ij}=\frac{\sum_ {i,j:i\neq j}E_{ij}}{\sum_ {i,j}d_id_j}d_id_j$. The proposed estimator is $\hat{p}_ {ij}=1\wedge \frac{\sum_ {i,j:i\neq j}E_{ij}}{\sum_ {i,j:i\neq j}d_id_j}d_id_j$. Discounting the clipping on $1$, which one can always do knowing that the matrix to be estimated has entries below $1$, the only difference is the factor $\sum_ {i,j:i\neq j}d_id_j$ is the denominator of $\hat{p}_ {ij}$, compared to $\sum_ {i,j}d_id_j$ in $p'_ {ij}$. In the dense regime treated here, I believe that the difference is negligible.I suspect **Algorithm 2** could be similarly treated. The power iteration is known to converge in one iteration for matrices of rank 1 and fast for low rank matrices. Comparing with this approach may be beneficial, as the direct competitors here are spectral methods, and it’s possible the methods are analogous.

2- **Difficulty in Extending Results**: Although the authors discuss potential extensions, constructing estimators for lower-rank graphons appears challenging as the equations grow complex quickly. Furthermore, the order (in powers of $E$) of the quantities needed increases with the rank, complicating the computation. Although the subsampling technique seems feasible, it lacks a thorough theoretical foundation. For example, in **Remark 2**, the justification only considers asymptotics, and I suspect finite sample considerations could introduce additional variance. Providing more detail on possible extensions would add value.

3- **Somewhat Unfair Experimental Comparisons**: In their experiments, the authors compare their method to more generic methods that work under broader assumptions (not strictly rank 1 or 2). For example, **USVT** can adapt to various low-rank situations. I consider this comparison somewhat unfair, as the proposed method is tailored to specific ranks in the application, effectively using additional information. Additionally, the authors note that methods are used with default values. Spectral methods with low-rank information might perform as efficiently as the proposed method.

4- **Limited Applicability**: Limiting the methodology to rank-1 and rank-2 graphons restricts its applicability. Additionally, for rank-2 graphons, **Assumption 1** feels restrictive. For instance, the condition $\int^1_0G_k(u)du\neq 0$, for all $k$, is unmet for $ k = 2 $ when \$ G_1 $ is constant (given $L_2 $ orthogonality). It seems natural to consider polynomial bases as the elements $ G_k $ in a low-degree graphon, which will typically include a constant function.

**Questions:**

1- Could you address my comments in the weaknesses section above, point by point?

2- The notation $O_p$ used throughout the paper is not defined.

3- In the finite sample results, such as in Theorem 1, the results should hold with high probability—this is currently unstated.

4- In line 434, the authors mention that the absence of a tuning parameter "enhances robustness." Could they provide a clearer justification for this claim?

5- f I am following their proof correctly, it seems that alternative equations involving $\hat{\lambda}_ 1$ and $\hat{\lambda}_ 2$ could be derived for the rank-2 case.  Specifically, it seems that $\sum_{i,j}A_ {ij}A_{ji}$ should approximate up to rescaling $\hat{\lambda}^2_1+\hat{\lambda}^2_2$. Is this correct, and would it be useful? How does this approach compare with the method they propose?

---

> ### Author Response · Authors · 2024-11-18
> **Response letter to Reviewer 3djw (part 1)**
>
> Thank you for reviewing our manuscript and providing valuable feedback. It is encouraging that you could acknowledge the clarity, theoretical foundation, and simulation results of our paper.  We summarize the main changes according to the valuable suggestions from reviewers as follows.
>
> 1.  We provide a complete theoretical framework for general $r$, updated in Lines 203–338 on pages 4–7.
> 2. Computationally, we introduce a new approximation approach for computations, updated in Lines 339–381 on pages 7–8. None of the theoretical results are affected, while it achieves a time complexity matching that of matrix multiplication ($ O(n^{2.373})$), eliminating the need for subsampling techniques.
> 3. In Appendix A.3, we provide a method for selecting $r$ when it is unknown.  We also point out this in Remark 2 of the main article.
> 4. We update the simulation results and included an additional simulation for $ r = 3 $.
> 5. In Appendix A.1, we include a real data analysis, demonstrating the use of our method for selecting $r$, as well as our algorithms for estimating the connection matrix and the graphon function.
> 6. We update the mathematical proofs for all theoretical results.
>
> In response to the weaknesses and questions you pointed out, we have made some additions and clarifications. Below, we address your questions point by point.
>
> - *``Regarding the proposed model Additive Separable Graphons ASG(r), the authors state in line 076 that this model is new. However, it appears to me as a generic low-rank graphon, which is a well-recognized particular case of a general graphon already discussed in the literature (e.g., Chan and Airoldi, 2014; Xu, 2018). Indeed, for any rank-r graphon, the decomposition given in Eq. (1)  follows directly from the spectral theorem. I struggle to see this as a new model, and in my opinion, it would be beneficial for the authors to justify why ASG(r) is indeed novel.''*
>
> Answer: Thank you for your question. As you noted, our model has appeared in other contexts and represents a natural formulation of low-rank models. However, this work is likely the first to leverage this model specifically for estimating both the connection probability matrix $P$ and, more importantly, the graphon function $f(u,v)$.
>
> While USVT can partially utilize low-rank information to estimate the connection probability matrix, it cannot estimate the graphon function over the full domain $[0,1] \times [0,1]$. It is worth noting that the connection probability matrix is derived by evaluating the graphon function at a finite set of discrete points. In contrast, our model allows direct estimation of the graphon function, leveraging its low-rank structure in a way that distinguishes our approach.
>
> Moreover, our method is tuning-parameter-free, making it computationally faster in practice compared to existing methods that require tuning parameters for graphon estimation. For further discussion, please refer to the third point below.
>
> - *``The power iteration is known to converge in one iteration for matrices of rank 1 and fast for low-rank matrices. Comparing with this approach may be beneficial, as the direct competitors here are spectral methods, and it’s possible the methods are analogous.''*
>
> Answer: Thank you for your insightful comments! The connection between our method and spectral methods is indeed an interesting topic. When $r = 1$, the results from the spectral method closely align with those of our method, as shown in Table 5 on page 14.
>
> Regarding power iteration, we would like to point out that although the connection matrix $P$ is rank-1, the corresponding adjacency matrix $E$ may still be full-rank. Additionally, both the spectral method and the power iteration method face challenges when extended to estimating graphon functions over $[0,1]$.
>
> For $r > 1$, the relationship between these methods and ours remains unclear. We have not identified any definitive connections, either through theoretical analysis or simulations. Moreover, our motivations differ fundamentally, as our goal is to estimate the graphon function itself, rather than just the connection probability matrix. It is important to note that the graphon function captures the core of the statistical model, while the connection matrix is merely a realization of the graphon function evaluated at a finite set of discrete points. Furthermore, the connection probability matrix is not a proper limit object in random graph models, whereas the key advantages of graphon models arise precisely in this limiting sense.
>
> From a practical perspective, our method tends to be slightly faster than USVT, as implemented in the R package provided by the authors of USVT.

---

> > ### Comment · Reviewer_3djw · 2024-11-22
> > **Discussions on your response (part 1)**
> >
> > I would like to thank the authors for their point-by-point response.
> >
> > **Regarding the novelty of the model**  The authors state, "However, this work is likely the first to leverage this model specifically for estimating both the connection probability matrix and, more importantly, the graphon function." However, this argument pertains more to the methods applied rather than the novelty of the model itself, and therefore, it is not entirely convincing in my opinion. Statements such as "...we introduce the additive separable model as a new, parsimonious representation of the graphon..." or "...introduces our new low-rank graphon model, termed the Additive Separable Graphon Model (ASG)..." seem, in my view, to overstate the novelty of the model. These claims could benefit from greater precision to avoid suggesting that the model itself is entirely new when it appears to build on established concepts.
> >
> > **Regarding the power iteration** The authors note, "we would like to point out that although the connection matrix is rank-1, the corresponding adjacency matrix may still be full-rank." While this observation is valid, I believe the conclusion should hold approximately, up to noise terms. Specifically, for a rank-1 $P$, the adjacency matrix for sufficiently large $n$ is expected to exhibit a dominant eigenvalue, with the remaining eigenvalues falling below a certain threshold.
> >
> > **Regarding the emphasis on graphon function estimation** The authors state, "our motivations differ fundamentally, as our goal is to estimate the graphon function itself, rather than just the connection probability matrix." However, this distinction is not consistently clear in the manuscript. While the estimation of the graphon function is indeed highlighted as a key contribution (and, in my opinion, the most interesting aspect of the work), the estimation of the connection probability matrix is also prominently presented as a main contribution.

---

> > > ### Author Response · Authors · 2024-11-26
> > > **Replies to Reviewer 3djw**
> > >
> > > Thank you for discussions. Based on your discussions, we have made some additions and modifications. Below, we address your questions point by point.
> > >
> > > - *''Statements such as ''...we introduce the additive separable model as a new, parsimonious representation of the graphon..." or ``...introduces our new low-rank graphon model, termed the Additive Separable Graphon Model (ASG)..." seem, in my view, to overstate the novelty of the model. These claims could benefit from greater precision to avoid suggesting that the model itself is entirely new when it appears to build on established concepts.''*
> > >
> > > Answer: Thank you for your valuable suggestions. We have revised the wording to reduce the emphasis on the novelty of the model itself. Instead, we frame our estimation approach as a new attempt to estimate a low-rank representation. Specifically, in line 11, we rephrased the sentence from "… as a new, parsimonious representation of the graphon" to "… as a parsimonious representation of the graphon." Similarly, in line 78, we modified the sentence from "Section 2 introduces our new, parsimonious low-rank graphon model" to "Section 2 introduces our parsimonious low-rank graphon model." Additionally, in line 108, we revised "we propose a new, parsimonious model" to "we propose a parsimonious model." Once again, thank you for your insightful feedback.
> > >
> > > - *``While this observation is valid, I believe the conclusion should hold approximately, up to noise terms. Specifically, for a rank-1 $P$, the adjacency matrix  for sufficiently large $n$ is expected to exhibit a dominant eigenvalue, with the remaining eigenvalues falling below a certain threshold.''*
> > >
> > > Answer: Thank you for your insightful suggestion. In response, we have added a new remark (Remark 1) that discusses the power iteration method and included it in our simulation studies (see Tables 2, 3, and 5). Our results show that while this method produces similar estimates for dense graphons, it performs less effectively for sparse ones.
> > >
> > > - *``The authors state, ''our motivations differ fundamentally, as our goal is to estimate the graphon function itself, rather than just the connection probability matrix." However, this distinction is not consistently clear in the manuscript. While the estimation of the graphon function is indeed highlighted as a key contribution (and, in my opinion, the most interesting aspect of the work), the estimation of the connection probability matrix is also prominently presented as a main contribution.''*
> > >
> > > Answer: Thank you for your suggestion. Apart from highlighting the novelty of estimating the graphon, as noted in line 55 of the article, we have also listed the estimation of the connection probability matrix as a distinct contribution, as explained in Remark 3. Specifically, the key advantage of our method is its ability to estimate both the graphon and the connection probability matrix simultaneously. From an algorithmic perspective, it differs from existing methods, and in terms of interpretation, it provides readers with a fresh and valuable perspective.
> > >
> > > - *``While it is commendable that the authors have extended their results, the distinction between their approach and spectral methods, particularly for estimating the probability matrix, remains somewhat unclear. Given that the complexity of the proposed method is of the order of matrix multiplication, it appears comparable to that of spectral methods.''*
> > >
> > > Answer: Thank you for your question. In response, we have added Remark 3 (line 289) in the revised manuscript to further highlight the key differences between our method and spectral methods, including differences in motivation, estimation procedure, assumptions, and empirical performance for sparse graphons.
> > >
> > > - *``Furthermore, leveraging the additional information of low-rank or approximate low-rank structure (ignoring noise terms), one might expect that methods like power iteration, with a constant number of iterations, could achieve similar results at a comparable complexity. Clarifying how the proposed method diverges from or improves upon these established techniques would help delineate its advantages more effectively.''*
> > >
> > > Answer: Thank you for your suggestion. We have observed that the power iteration method performs similarly to our approach for dense graphons, but underperforms for sparse ones. For more details, please refer to Remark 1. We would also like to emphasize that, unlike these numerical-based methods, our approach is fundamentally rooted in statistical principles. It not only estimates the graphon itself but also provides an estimate of the graphon evaluated at the observed nodes.

---

> ### Author Response · Authors · 2024-11-18
> **Response letter to Reviewer 3djw (part 2)**
>
> - *``Difficulty in Extending Results: Although the authors discuss potential extensions, constructing estimators for lower-rank graphons appears challenging as the equations grow complex quickly. Furthermore, the order (in powers of E) of the quantities needed increases with the rank, complicating the computation.''*
>
> Answer: Thank you for your suggestion. We have now extended the model to arbitrary $r$, provided $r$ is bounded. The theoretical complexity of our revised model remains $O(n^{2.373})$ (see Remark 3). Our simulation results also confirm that the proposed method is computationally efficient in practice (see Table 2). Additionally, we have conducted a thorough theoretical analysis for the case $r > 2$, which is detailed in Lines 203–338 on pages 4–7 of the revised version. In this more general setting, we obtain the same consistent results as before.
>
> - *``Although the subsampling technique seems feasible, it lacks a thorough theoretical foundation. For example, in Remark 2, the justification only considers asymptotics, and I suspect finite sample considerations could introduce additional variance. ''*
>
> Answer: Thank you for your suggestions, which have significantly enhanced our work. After we updated the theory for general $r$, we find a new approach for computing to replace the original sampling methods in Lines 338-381 on pages 7-8. Specifically, we approximate simple paths with paths allowing node repetition and prove their asymptotic equivalence, ensuring that this approximation does not affect any theoretical results. This approach enables our method to achieve a time complexity matching that of matrix multiplication ($O(n^{2.373})$), eliminating the need for subsampling techniques. In practice, we also suggest eliminating paths with repetitive nodes subject to the computational constraint to improve the finite sample performance. In fact, when $r$ is small, the extra computational cost of this correction can be small, as confirmed by our simulation. Additionally, we have included a simulation result in section 4 for $r=3$. With this new approach, subsampling is no longer required. Once again, thank you for your insightful question, as it motivated us to develop this new method.
>
> - *``Somewhat Unfair Experimental Comparisons: In their experiments, the authors compare their method to more generic methods that work under broader assumptions (not strictly rank 1 or 2). For example, USVT can adapt to various low-rank situations. I consider this comparison somewhat unfair, as the proposed method is tailored to specific ranks in the application, effectively using additional information. Additionally, the authors note that methods are used with default values. Spectral methods with low-rank information might perform as efficiently as the proposed method.''*
>
> Answer:  Thank you for your suggestion. We appreciate the reviewer's insightful comment regarding the fairness of the comparison between methods. We conducted additional simulations where we used the true $r$ for USVT (i.e., retaining only the first $r$ eigenpairs). We report the results in Table 2 on page 9. Overall, the conclusion remains, namely, that our method performs on par with USVT for estimating the connection probability matrix. This is also expected, as both methods achieve the minimax optimal for estimating the connection probability matrix. Notably, our method offers a slight advantage in computational speed. Finally, we would like to emphasize that our method is capable of estimating the graphon function $f(u,v)$ over $[0,1]\times [0,1]$, and USVT cannot estimate the graphon function. We also compare with other estimating methods for the graphon function, such as the network histogram method proposed by Olhede and Wolfe, 2014. The network histogram algorithm relies on a greedy search to find the maximum likelihood estimator, leading to both high theoretical computational complexity and significant practical computational cost. Furthermore, our algorithm demonstrates superior accuracy with respect to both error metrics compared to the network histogram method, as shown in Table 2.
>
> - *``Limited Applicability: Limiting the methodology to rank-1 and rank-2 graphons restricts its applicability. Additionally, for rank-2 graphons, Assumption 1 feels restrictive.''*
>
> Answer: Thank you for your nice comments! In the revision, we have extended our method to address the model with a general $r$. For  Assumption 1, admittedly, our condition rules out $\int_0^1G_k(u)du=0$ for some $k$. We note that similar assumptions has made in other works, such as in Theorem 3 of [1]. Exploring how to relax this condition is an important and influential future work.
>
> [1] Bickel, P. J., Chen, A., & Levina, E. (2011). The method of moments and degree distributions for network models. The Annals of Statistics, 39(5), 2280–2301.

---

> > ### Comment · Reviewer_3djw · 2024-11-22
> > **Discussion on your response (part 2)**
> >
> > **Regarding the Extension of the Results and the Complexity** While it is commendable that the authors have extended their results, the distinction between their approach and spectral methods, particularly for estimating the probability matrix, remains somewhat unclear. Given that the complexity of the proposed method is of the order of matrix multiplication, it appears comparable to that of spectral methods.
> >
> > Furthermore, leveraging the additional information of low-rank or approximate low-rank structure (ignoring noise terms), one might expect that methods like power iteration, with a constant number of iterations, could achieve similar results at a comparable complexity. Clarifying how the proposed method diverges from or improves upon these established techniques would help delineate its advantages more effectively.

---

> ### Author Response · Authors · 2024-11-18
> **Response letter to Reviewer 3djw (part 3)**
>
> - *``The notation  $O_p$ used throughout the paper is not defined. In the finite sample results, such as in Theorem 1, the results should hold with high probability—this is currently unstated.''*
>
> Answer: Thank you for pointing this out. We have now included the definition of $O_p$ in Section 1 in Line 88 on page 2. After properly defining $O_p$, the statement of Theorem 1 inherently includes the meaning of ``hold with high probability''.
>
> - *``In line 434, the authors mention that the absence of a tuning parameter "enhances robustness." Could they provide a clearer justification for this claim?''*
>
> Answer: For nonparametric methods (e.g., the SAS method by Chan and Airoldi, 2014, and the network histogram method by Olhede and Wolfe, 2014), a bandwidth parameter $h$ (or $K=n/h$) is required, which affects estimation accuracy and can be challenging to select in practice. In our simulations, we used their suggested default value, which achieves the optimal theoretical convergence rate. Additionally, we conducted extra simulations and found that choosing different values of their parameter results in fluctuations in the MSE, see two tables below. For instance, increasing the  $K$ for the SAS method to 1.2 times the default value led to an approximately 1.6-fold increase in MSE.  Similarly, increasing the bandwidth $h$ of the network histogram method by a factor of 0.5 leads to a twofold increase in the MSE.  This indeed highlights the importance and difficulty of parameter selection in practice. In contrast, our method does not require parameter tuning, eliminating this complication.
>
> ### The MSE of the SAS (Chan and Airoldi, 2014) under different parameter selections.
> | K   | 190   | 210   | 230   | 250 (default)   | 270   | 290   | 310   |
> |-----|-------|-------|-------|-------|-------|-------|-------|
> | MSE | 0.00155 | 0.00192 | 0.00235 | 0.00283 | 0.00340 | 0.00398 | 0.00462 |
>
> ### The MSE of the Nethist (Olhede and Wolfe, 2014) under different parameter selections.
> | h     | 30    | 40    | 50 (default)   | 60    | 70      |
> |------|-------|-------|-------|-------|------- |
> | MSE |  0.000873 | 0.000713 | 0.000617 | 0.000582 | 0.000525  |
>
> - *``I am following their proof correctly, it seems that alternative equations involving $\hat\lambda_1$ and $\hat\lambda_2$ could be derived for the rank-2 case. Specifically, it seems that $\sum_{i,j}A_{ij}A_{ji}$ should approximate up to rescaling $\lambda_1^2+\lambda_2^2$. Is this correct, and would it be useful? How does this approach compare with the method they propose?''*
>
> Answer: Thank you for your interest in the specifics. We kindly recall that, in this paper, we consider simple undirected graphs for modeling the network (i.e., without multiple edges or self-loops, and $A_{ij}=A_{ji}$). Consequently, we have $\sum_{i,j}A_{ij}A_{ji}=\sum_{i,j}A_{ij}$ which approximate up to rescaling $\lambda_1(\int_0^1 G_1(u)du)^2+\lambda_2(\int_0^1 G_2(u)du)^2$. We propose new methods instead of the previous sampling approach, for effectively estimating ASG(r) model utilizing lines and cycles (see Lines 223–227 on page 5 for a formal definition). We choose these subgraphs because they not only ensure theoretical guarantees (Theorems 3 and 4) but also maintain computational efficiency. As demonstrated in Section 3.3, our method achieves an asymptotically equivalent complexity to matrix multiplication, effectively avoiding the challenges of general subgraph counting. In practice, we also suggest eliminating paths with repetitive nodes subject to the computational constraint to improve the finite sample performance. In fact, when $r$ is small, the extra computational cost of this correction can be small, as confirmed by our simulation (see the running time in Table 2 on page 9).

---

> > ### Comment · Reviewer_3djw · 2024-11-22
> > **Discussion of your response (part 3)**
> >
> > **Regarding Alternative Equations for Eigenvalue Estimation** The newly proposed method adequately addresses my earlier question. However, the cycles appearing in equation (5) should, in principle, be expressible in terms of the spectrum of the graph's adjacency matrix. Considering this and my prior comment regarding the complexity, I find it challenging to identify a clear advantage over spectral methods, particularly for the estimation of $P$. Further elaboration on the specific improvements or distinct benefits of the proposed approach in this context would strengthen its positioning.

---

> > > ### Author Response · Authors · 2024-11-26
> > > **Replies to Reviewer 3djw (Continued)**
> > >
> > > - *``However, the cycles appearing in equation (5) should, in principle, be expressible in terms of the spectrum of the graph's adjacency matrix. Considering this and my prior comment regarding the complexity, I find it challenging to identify a clear advantage over spectral methods, particularly for the estimation of $P$. Further elaboration on the specific improvements or distinct benefits of the proposed approach in this context would strengthen its positioning.''*
> > >
> > > Answer: Thank you for your suggestion. The differences between our method and spectral methods are highlighted in our previous response. Notably, our approach allows for the simultaneous estimation of both the connection probability matrix and the graphon function. Additionally, for estimating the connection probability matrix, our method shows clear advantages over spectral methods in the sparse region (see Table 3). Theoretically, even for Lipschitz graphons, spectral methods do not achieve optimal rates in sparse settings (see Xu, 2018). Exploring the convergence rate of our method in this sparse region presents a promising direction for future work.
> > >
> > > We sincerely appreciate your suggestion to consider the power iteration and spectral methods, which prompted us to further investigate their empirical performance in this revision.

---

> > > > ### Comment · Reviewer_3djw · 2024-11-28
> > > >
> > > > I thank the authors for their response and discussions. I believe the modification they introduce have improved their paper.
> > > >
> > > > In light of this new version, I have a couple of extra comments: please review that you are not using 'new model'. I found a one instance of this in line 48. I think removing all this give a more transparent view on this model.
> > > >
> > > > Additionally, could the authors clarify how they use the power iteration in the following two respects:
> > > > 1) How many iterations did they use?
> > > > 2) How they estimate the eigenvectors for ranks larger than one? Did they use the deflation technique?
> > > > 3) Did they apply the same clipping to 1 in the other estimators they tested, apart from the one they proposed? In my initial comment, I pointed out: "Discounting the clipping on 1, which one can always do knowing that the matrix to be estimated has entries below 1". Given that their proposed estimator explicitly leverages this additional information (that the matrix has entries in $[0,1]$), it seems that any general-purpose algorithm could also be adapted to incorporate this knowledge.

---

> > > > > ### Author Response · Authors · 2024-11-29
> > > > >
> > > > > Dear reviewer 3djw,
> > > > >
> > > > > &nbsp;
> > > > >
> > > > > ---
> > > > >
> > > > > Thank you for your response. Based on your feedback, we have addressed your questions point by point, as outlined below.
> > > > >
> > > > > General question: *``please review that you are not using 'new model'. I found a one instance of this in line 48. I think removing all this give a more transparent view on this model.''*
> > > > >
> > > > > Answer: Thank you very much for your suggestion. We have corrected this issue, deleting all the terms 'new model' in the article. This correction will be fully reflected in the final version of the submission. To highlight and distinguish the contributions of our work, we have decided to add the following: "While low-rank decomposition is a well-established concept, our approach is novel in its use of this decomposition to iteratively and efficiently estimate eigenpairs. We leverage network moments throughout the process, providing an alternative computationally efficient framework that builds on this structure in a way not previously explored. Notably, our framework can be applied to estimate the graphon function itself, in addition to the connection probability matrix that low-rank decomposition and spectral methods focus on."
> > > > >
> > > > > - *``How many iterations did they use?''*
> > > > >
> > > > > Answer:  For power iteration, we set the maximum number of iterations to $500$ and the convergence threshold to $10^{-6}$ (measured by the norm of the difference between successive vectors, see the algorithm in the response to the next question). For all scenarios discussed in our paper, the actual number of iterations is listed in the tables below, and we have added the tables in the appendix.
> > > > >
> > > > > The results indicate that in most scenarios, the number of iterations required for convergence is relatively small. However, there are exceptions in which the power iteration does not achieve convergence within 500 iterations.
> > > > >
> > > > > | ID  | $\hat \lambda_1$ iteration | Std.Dev. | $\hat \lambda_2$ iteration | Std.Dev. | $\hat \lambda_3$ iteration | Std.Dev. |
> > > > > |:---:|:--------------------------:|:--------:|:--------------------------:|:--------:|:--------------------------:|:--------:|
> > > > > | 1   | 6                          | 0        |                            |          |                            |          |
> > > > > | 2   | 5                          | 0        |                            |          |                            |          |
> > > > > | 3   | 5.01                       | 0.0995   |                            |          |                            |          |
> > > > > | 4   | 9                          | 0        | 17.99                      | 0.5744   |                            |          |
> > > > > | 5   | 6                          | 0        | 500                        | 0        |                            |          |
> > > > > | 6   | 15.34                      | 0.6200   | 8                          | 0        |                            |          |
> > > > > | 7   | 27.83                      | 1.8871   | 15.1                       | 0.7416   | 8                          | 0        |
> > > > >
> > > > > **Table1:** The number of iterations for dense settings across 100 independent trials.
> > > > >
> > > > >
> > > > > | ID  | $\hat \lambda_1$ iteration | Std.Dev. | $\hat \lambda_2$ iteration | Std.Dev. |
> > > > > |:---:|:--------------------------:|:--------:|:--------------------------:|:--------:|
> > > > > | 2   | 13                         | 0        |                            |          |
> > > > > | 3   | 17.15                      | 0.3571   |                            |          |
> > > > > | 4   | 31.91                      | 1.8713   | 500                        | 0        |
> > > > > | 5   | 17.98                      | 0.3995   | 500                        | 0        |
> > > > >
> > > > > **Table2:** The number of iterations for sparse settings across 100 independent trials.

---

> ### Author Response · Authors · 2024-11-29
> **Continued**
>
> - *``How they estimate the eigenvectors for ranks larger than one? Did they use the deflation technique?''*
>
> Answer: Thank you for your question. We confirm that we have employed the deflation technique, as outlined in Stoer et al. (1980). To clarify this approach, we have added the following remark in the appendix, where we describe how we estimate the eigenpairs for ranks greater than one, with a particular emphasis on the deflation technique.
>
> ``Suppose we want to estimate the leading $r$ eigenpairs of a matrix $A$. The power iteration algorithm can be used by iteratively applying the power method and deflating the matrix. Beginning with $A_0 = A$, each step $k$ (from $1$ to $r$) involves estimating the dominant eigenpair $(\hat \lambda_k, \hat v_k)$ of $A_{k-1}$ using the power iteration method (with normalization of $\hat v_k$) and then deflating the matrix via $A_k = A_{k-1} - \hat \lambda_k \hat v_k \hat v_k^\top$. After $r$ steps, the estimated eigenpairs are used to reconstruct an approximation $\tilde P = \sum_{k=1}^r \hat \lambda_k \hat v_k \hat v_k^\top$. Finally, the estimated probability connection matrix $\hat P$ is obtained by applying an element-wise thresholding operation to $\tilde P: \hat p_{ij} = 1 \wedge (0 \vee \tilde p_{ij})$, ensuring entries fall within the $[0, 1]$ probability range. The whole algorithm is
>
> **Algorithm:** Power Iteration for $ASG(r)$.
>
> **Input:** adjacency matrix $A$, rank $r$, maximum number of iterations $N$, tolerance $\epsilon$
>
> **Output:** estimated probability connection matrix $\hat{P}$.
>
> 1. Initialize $A_0 \gets A$.
>
> 2. For $k = 1$ to $r$:
>
> 3. Initialize vector $\mathbf x_0$ of length $n$ with all ones. Normalize $\mathbf x_0 \gets \frac{\mathbf x_0}{\|\mathbf x_0\|_2}$.
>
> 4. For $i = 1$ to $N$:
>
> 5. $\mathbf x_{i} \gets A_{k-1} \mathbf x_{i-1}$.
>
> 6. Normalize $\mathbf x_{i} \gets \frac{\mathbf {x_{i}} }{\|\mathbf x_{i}\|_2}$.
>
> 7. If $\|\mathbf x_{i} - \mathbf x_{i-1}\|_2 < \epsilon$: break; End If
>
> 8. End For
>
> 9. With deflation technique, compute eigenvalue $\hat \lambda_{k} \gets \mathbf x_i^T A_{k-1} \mathbf x_i$. Set eigenvector $\mathbf{\hat v_k} \gets \mathbf{x_i}$. Update $A_k \gets A_{k-1} - \hat \lambda_k \mathbf{\hat{v}_k} \mathbf {\hat{v}_k}^\top$.
>
> 10. End For
>
> 11. Compute $\tilde P = \sum_{k=1}^r \hat \lambda_k \mathbf{\hat v_k} \mathbf{\hat v_k}^\top$, and take $\hat p_{ij} = 1 \wedge (0 \vee \tilde P_{ij})$ for $1\le i\neq j\le n, \hat p_{ii}=0$ for $1\le i\le n$.
>
> 12. Return estimated probability connection matrix $\hat P =( \{\hat p_{ij}\})_{1 \leq i, j \leq n}$.
> ''
>
> Thank you again for your suggestion. We will include these statements in the appendix of the final version.
>
> - *``Did they apply the same clipping to 1 in the other estimators they tested, apart from the one they proposed? In my initial comment, I pointed out: "Discounting the clipping on 1, which one can always do knowing that the matrix to be estimated has entries below 1". Given that their proposed estimator explicitly leverages this additional information (that the matrix has entries in $[0,1]$), it seems that any general-purpose algorithm could also be adapted to incorporate this knowledge.''*
>
> Answer: Thank you for your question. We would like to clarify that in our current version, all the methods (both ours and others') include this adjustment, i.e., using $1 \wedge (0 \vee \hat{p}_{ij})$ to calculate the final MSE and maximum error. We have added the statement at the beginning of Section 4.

---

> > ### Comment · Reviewer_3djw · 2024-12-01
> >
> > I would like to thank the authors once again for their response. I believe the introduced modifications have enhanced the scope of their results. Out of curiosity, I have a final question: do the authors have any insights into why their method appears to experimentally outperform competing methods in the sparse regime?

---

> > > ### Author Response · Authors · 2024-12-01
> > >
> > > Thank you for your question. Our understanding is informed by Theorem 1 in Bickel et al. (2011), which establishes favorable convergence rates for the method of moments estimator, particularly in sparse settings. Our estimator builds on this foundation by leveraging specific network moments, although the precise details of its convergence properties require further investigation. Given the scope of this paper and time constraints, we propose leaving a deeper exploration of this topic for future research.

---

> > > > ### Comment · Reviewer_3djw · 2024-12-01
> > > >
> > > > Thank you for your response and for providing the reference. Indeed, a deeper exploration of this topic could be a valuable direction for future work. I would suggest adding this insight as a comment in the paper, if it has not been included already.
> > > >
> > > > Beyond this, I believe the revisions have improved the paper, and I have updated my score accordingly.

---

> > > > > ### Author Response · Authors · 2024-12-01
> > > > >
> > > > > Thank you very much! We have added this insight as a comment to the final version of the paper.

---

### Meta-Review · Area_Chair_6xEE · 2024-12-20

**Metareview:**

This paper studies algorithms for estimating low-rank graphons. They first estimate the low-rank probability matrix for the finite node sample and then use linear interpolation to estimate the full graphon function on [0,1]->[0,1]. Reviewers were generally positive about this paper, feeling that the model studied was nice and simple, the writing is clear and rigorous, and the experimental results are convincing.

However, after discussion, we felt that 1) some substantial aspects still need further exploration/improvement and 2) given the substantial changes made during the rebuttal the paper should really see a resubmission and re-review in its new form. A summary of these issues/changes is below:

1. The initial version claimed that the low-rank graphon model was new and introduced in this paper. This is inaccurate -- low-rank random graph models and graphons have appeared in significant prior work. The authors edited the paper to clarify that the model was not novel, although some of the language is still a bit vague.
2. The initial paper had limited scope in that it only gave theoretical results for rank <= 2 graphons. During the rebuttal period, the authors  substantially strengthened the results to capture rank r graphons for any r and the main algorithm (Algorithm 2) has changed substantially. This is a very positive outcome of the rebuttal period, however, the changes warrant a new review of the paper for correctness.
3. As one reviewer pointed out, the paper's approach for estimating the edge probability matrix is very close to a spectral method: it looks like one iteration of power method for estimating the top eigenvector of the adjacency matrix. While the authors have added some discussion of spectral methods, we feel that before publication, a better understanding of how these methods compare is really needed. Theoretically, how would results change if one instead just computed the top eigenvector of the adjacency matrix (or top r eigenvectors in the rank-r case)  to estimate the connection probability matrix? It does not feel enough to simply mention that the method is similar to spectral methods without exploring this further. The authors point out that spectral methods only estimate the finite-sized connection probability matrix, and not the full graphon function on [0,1] x [0,1]. But this was not a very convincing argument: after all, the two main algorithms presented in this work also focus on estimating the connection probability matrix, and the graphon is then estimated through interpolation. Interpolation could be applied to any method that first estimates the finite connection probability matrix.

On balance, we feel that given the above issues, the paper warrants a resubmission/re-review.

**Additional Comments On Reviewer Discussion:**

See main meta review.

---

> ### Public Comment · ~Xinyuan_Fan1 · 2025-02-13
>
> We regret being rejected with a score of 666—reflecting all positive scores—after substantial improvements. We fully respect the decision, however, there are some points we need to clarify:
>
> 1. During the rebuttal period, we extended our work to cover cases with arbitrary fixed $r$. The algorithms and theory are, of course, firmly grounded in the methods and theoretical framework originally submitted for $r\le 2$. Therefore, the correctness is guaranteed. Furthermore, the reviewers acknowledged this extension without raising any concerns about its correctness. Finally, considering that the ICLR website clearly states that "reviewers are not required to read the appendix", I have reservations about the statement "the changes warrant a new review of the paper for correctness".
>
> 2. Our method and the spectral method differ significantly in motivation, assumptions, and performance in sparse scenarios, as clearly laid out in Remark 3. Moreover, recovering the graphon function from an estimate of the connection probability matrix is, indeed, far from trivial.  First, the latent variables associated with the graphon are both unknown and unordered, which prevents the connection probability matrix from being treated as a lattice sampling of the underlying graphon. Second, further analyzing the estimated connection probability matrix becomes complex, particularly when the goal is to achieve sup-norm consistency for graphon estimation. It is truly a pity that there are still misunderstandings regarding this problem, and, somehow, regarding our methods and theoretical framework.

---

### Decision · Program_Chairs · 2025-01-22

Reject